# Disruption of FOXO3a-miRNA feedback inhibition of IGF2/IGF-1R/IRS1 signaling confers Herceptin resistance in HER2-positive breast cancer

Liyun Luo[1,4], Zhijie Zhang[1,4], Ni Qiu[1,4], Li Ling[1], Xiaoting Jia[1], Ying Song [1], Hongsheng Li[1], Jiansheng Li[1], Hui Lyu [2], Hao Liu [1], Zhimin He [1✉], Bolin Liu [2✉] & Guopei Zheng [1,3✉]

Resistance to Herceptin represents a significant challenge for successful treatment of HER2-positive breast cancer. Here, we show that in Herceptin-sensitive cells, FOXO3a regulates specific miRNAs to control IGF2 and IRS1 expression, retaining basic IGF2/IGF-1R/IRS1 signaling. The basic activity maintains expression of PPP3CB, a subunit of the serine/threonine-protein phosphatase 2B, to restrict FOXO3a phosphorylation (p-FOXO3a), inducing *IGF2*- and *IRS1*-targeting miRNAs. However, in Herceptin-resistant cells, p-FOXO3a levels are elevated due to transcriptional suppression of *PPP3CB*, disrupting the negative feedback inhibition loop formed by FOXO3a and the miRNAs, thereby upregulating IGF2 and IRS1. Moreover, we detect significantly increased IGF2 in blood and IRS1 in the tumors of breast cancer patients with poor response to Herceptin-containing regimens. Collectively, we demonstrate that the IGF2/IGF-1R/IRS1 signaling is aberrantly activated in Herceptin-resistant breast cancer via disruption of the FOXO3a-miRNA negative feedback inhibition. Such insights provide avenues to identify predictive biomarkers and effective strategies overcoming Herceptin resistance.

[1] Affiliated Cancer Hospital & Institute of Guangzhou Medical University, Guangzhou, Guangdong, China. [2] Department of Genetics, Stanley S. Scott Cancer Center, School of Medicine, Louisiana State University (LSU) Health Sciences Center, New Orleans, LA, USA. [3] Guangzhou Municipal and Guangdong Provincial Key Laboratory of Protein Modification and Degradation, Guangzhou, Guangdong, China. [4] These authors contributed equally: Liyun Luo, Zhijie Zhang, Ni Qiu. ✉email: hezhimin2005@yahoo.com; bliu2@lsuhsc.edu; zhengguopei@126.com

Breast cancer is the leading cause of cancer-related deaths in women worldwide[1]. As a heterogeneous disease, breast cancer has been classified into several molecular subtypes[2], including HER2-positive breast cancer which is defined as the subtype with amplified and/or overexpressed *HER2* (or *erbB2*) gene[3]. Amplification/overexpression of *HER2* is observed in approximately 20–25% of breast cancers and significantly associated with poor prognosis in breast cancer patients[3–5]. Herceptin (or trastuzumab), a humanized anti-HER2 monoclonal antibody (Ab), is an effective HER2-targeted therapy against early and metastatic HER2-positive breast cancers. It has dramatically improved survival of breast cancer patients with HER2-positive tumors[6,7]. However, not all HER2-positive breast cancers respond to Herceptin-based regimens. Majority of the patients who achieve an initial response become resistance within one year[8–10]. Resistant tumors likely recur and metastasize to distant organs, which accounts for approximately 90% of cancer deaths[4,11]. Many patients with advanced HER2-positive breast cancer ultimately develop brain metastasis[6]. Thus, both primary (de novo) and acquired resistances to Herceptin frequently occur and currently represent a significant clinical obstacle for successful treatment of HER2-positive breast cancer. To date, we lack validated biomarkers predictive for Herceptin response[7,12]. It is in urgent need to identify novel therapy overcoming Herceptin resistance with the aim to eliminate mortality of the patients with metastatic HER2-positive breast cancers.

Several mechanisms of Herceptin resistance in HER2-positive breast cancer have been proposed[6,10,13–15]. Among them, compensatory signaling activation by another receptor tyrosine kinase (RTK) attenuates Herceptin binding efficiency due to HER2 dimerization with the RTK. It also provides survival advantage to breast cancer cells, thereby resulting in resistance to Herceptin[6,10]. Herceptin resistance can also occur through mechanisms that lead to HER2 reactivation via acquisition of HER2 L755S mutation[16], activation of the PI-3K/Akt pathway via *PIK3CA* mutation[17] or *PTEN* loss[8], or extracellular matrix triggered integrin β1/Src activation[18]. In addition, a number of studies implicate various components of the insulin-like growth factor (IGF) system in breast cancer progression[19,20]. IGF1 and IGF2 are the major ligands in this system, and potent mitogens and anti-apoptotic peptides that affect cancer cell proliferation and survival via activation of the insulin-like growth factor-1 receptor (IGF-1R) signaling. Herceptin-induced growth inhibition was lost in breast cancer cells that overexpressed both HER2 and IGF-1R, and the growth arrest was regained when IGF binding protein-3 (IGFBP-3), which blocked IGF-induced IGF-1R signaling, was added[21]. However, the expression of IGF-1R per se did not predict Herceptin resistance in patients with HER2-positive breast cancer[22], suggesting that IGF-1R signaling via ligand-stimulation or interaction with another RTK was critical for the development of Herceptin resistance. Indeed, crosstalk occurred between IGF-1R and HER2, and IGF-1R physically interacted with HER2 and induced HER2 activation in Herceptin-resistant, but not -sensitive breast cancer cells[23]. Nonetheless, the precise mechanism through which IGF-1R signaling is highly activated in HER2-positive breast cancer resistant to Herceptin remains elusive. In this study, we seek to investigate the contributions of IGF2 and the insulin receptor substrate-1 (IRS1) to Herceptin resistance and elucidate the underlying mechanism of increased expression of both IGF2 and IRS1 and aberrant activation of IGF-1R signaling in Herceptin-resistant breast cancer.

## Results

**IGF2/IGF-1R/IRS1 signaling maintains Herceptin resistance phenotype.** Our previous studies showed that IGF-1R-initiated

signaling played an important role leading to Herceptin resistance in HER2-positive breast cancer cells[24]. To elucidate the underlying mechanism, we explored the regulation of IGF-1R signaling using Herceptin-resistant sublines SKBR3-pool2 (pool2) and BT474-HR20 (HR20), derived from SKBR3 and BT474, respectively, two well-known HER2-positive breast cancer cell lines sensitive to Herceptin[24]. Pool2 and HR20 cells as compared to SKBR3 and BT474, respectively, were resistant to Herceptin-mediated growth inhibition (Supplementary Fig. 1a). The resistance phenotype was confirmed in in vivo tumor xenografts models. While Herceptin significantly suppressed SKBR3-generated tumor growth in nude mice, it had little effect on the growth of tumors-established from pool2 cells (Supplementary Fig. 1b). Consistent with our previous findings[24], the protein levels of IGF-1R were similar in all cell lines (Fig. 1a). However, the levels of IRS1 and phosphorylated IGF-1R (p-IGF1R), Akt (p-Akt$^{(T308)}$ and p-Akt$^{(S473)}$), S6K (p-S6K), and FOXO3a (p-FOXO3a) were much higher in pool2 and HR20 cells than that in SKBR3 and BT474 cells, respectively (Fig. 1a). S6K is a downstream target of the mammalian target of rapamycin (mTOR) complex 1 (mTORC1)[25], whereas Akt$^{(S473)}$ and FOXO3a are downstream targets of mTOR complex 2 (mTORC2)[26,27]. The increase of p-Akt$^{(S473)}$, p-S6K, and p-FOXO3a suggested activation of both mTORC1 and mTORC2 in the resistant cells. To determine whether the ligands for IGF-1R might trigger activation of the signaling, we examined mRNA expression of *IGF1* and *IGF2* in the cells by quantitative real-time PCR (qRT-PCR) and measured the protein levels of IGF1 and IGF2 in the conditioned medium (CM) by ELISA. There was no significant difference of *IGF1* and *IGF2* mRNA between Herceptin-sensitive and -resistant cells (Supplementary Fig. 1c). However, we detected significantly higher protein levels of IGF2, but not IGF1 in the CM of pool2 and HR20 cells than that of SKBR3 and BT474 cells, respectively (Fig. 1b). These data suggest that post-transcriptional upregulation of IGF2 may play a role in the activation of IGF-1R/Akt/mTOR signaling in Herceptin-resistant breast cancer cells.

To investigate the importance of increased IRS1 and IGF2 in the IGF-1R/Akt/mTOR signaling activation and Herceptin resistance, we first utilized shRNAs to specifically downregulate IRS1 expression in pool2 and HR20 cells (Supplementary Fig. 1d). Specific knockdown of IRS1 significantly re-sensitized the resistant cells to Herceptin-mediated growth inhibition (Fig. 1c). We then performed *IRS1* gene deletion by CRISPR-Cas9 technology, which eliminated IRS1 expression in both pool2 and HR20 cells (Fig. 1e). Importantly, *IRS1* deletion also markedly re-sensitized the resistant cells to Herceptin treatment (Fig. 1d). Moreover, downregulation of IRS1 by either specific shRNAs or CRISPR-Cas9 gene editing clearly reduced the levels of p-IGF-1R, p-Akt$^{(T308)}$, p-Akt$^{(S473)}$, p-S6K, and p-FOXO3a in pool2 and HR20 cells (Fig. 1e). These data strongly supported that the expression of IRS1 was essential for the activation of IGF-1R/Akt/mTOR signaling. Consistently, specific inhibitors for IGF-1R (picropodophyllin, PPP), Akt (MK-2206), and mTOR (WAY-600) reversed the resistance phenotype of pool2 and HR20 cells (Supplementary Fig. 1e). In contrast, treatment of SKBR3 and BT474 cells with low dose (10 ng/ml) of recombinant human IGF2 (rhIGF2) elicited resistance to Herceptin in the otherwise sensitive cells (Supplementary Fig. 1f). Interestingly, high dose (80 ng/ml) of rhIGF2 (a similar IGF2 level as we detected in the CM of pool2 and HR20 cells (Fig. 1b)) alone slightly attenuated Herceptin-mediated inhibitory effect on SKBR3 and BT474 cells, whereas ectopic expression of IRS1 combined with high dose (80 ng/ml) of rhIGF2 significantly converted SKBR3 and BT474 into Herceptin-resistant cells (Fig. 1f). Collectively, our data indicated that the IGF-1R/Akt/mTOR signaling indeed played a crucial role resulting in Herceptin resistance. IGF2-induced activation of the signaling conferred Herceptin resistance, requiring IRS1 expression.

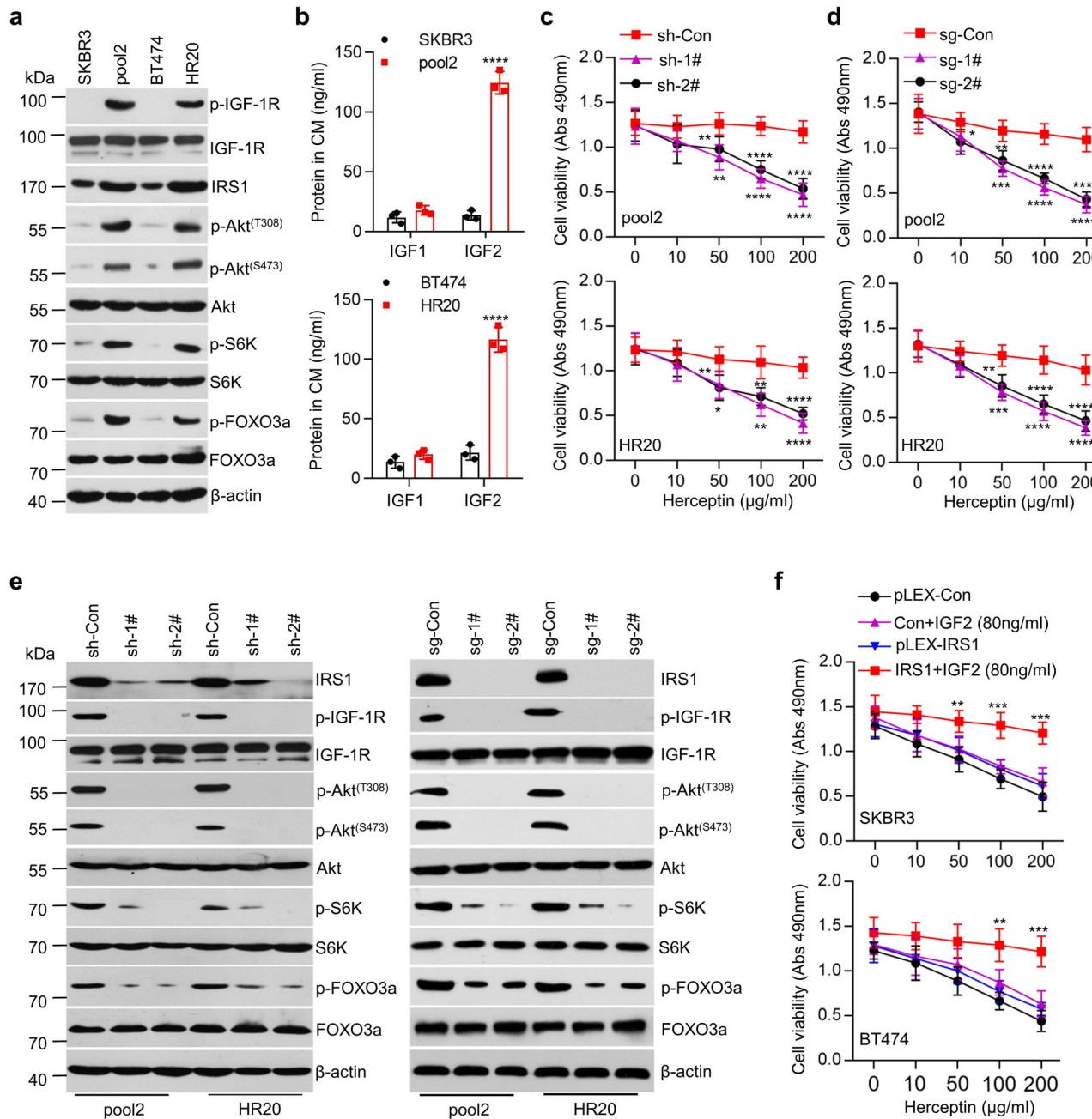

**Fig. 1 IGF2/IGF-1R/IRS1/Akt/mTOR signaling pathway was involved in resistance to Herceptin in HER2-positive breast cancer cells. a** SKBR3, pool2, BT474, and HR20 cells cultured at normal condition were collected and subjected to western blot analyses of p-IGF-1R, IGF-1R, IRS1, p-Akt(T308), p-Akt (S473), Akt, p-S6K, S6K, p-FOXO3a, FOXO3a, and β-actin. **b** The levels of IGF1 and IGF2 in conditioned medium (CM) were measured by ELISA, ****$p <$ 0.0001. **c** Pool2 or HR20 cells stably transfected with control shRNA (sh-Con) or *IRS1*-targeting shRNAs (sh-1#, sh-2#) were treated with Herceptin at indicated concentrations for 72 h. Cell viability was evaluated by MTS assays, Pool2 sh-1#: **$p = 0.0021$, ****$p < 0.0001$, sh-2#: **$p = 0.0096$, ****$p <$ 0.0001, HR20 sh-1#: *$p = 0.0154$, **$p = 0.0015$, ****$p < 0.0001$, sh-2#: **$p = 0.0072$, **$p = 0.0034$, ****$p < 0.0001$. **d** Pool2 or HR20 cells with *IRS1* deletion via CRISPR-Cas9 gene editing (sg-Con vs sg-1# and sg-2#) were treated with Herceptin at indicated concentrations for 72 h. Cell viability was evaluated by MTS assays, Pool2 sg-1#: ***$p = 0.0002$, ****$p < 0.0001$, sg-2#: *$p = 0.0235$, **$p = 0.0017$, ****$p < 0.0001$, HR20 sg-1#: ***$p = 0.0003$, ****$p < 0.0001$, sg-2#: **$p = 0.0023$, ****$p < 0.0001$. **e** Downregulation of IRS1 was achieved with either specific shRNAs (sh-Con vs sh-1# and sh-2#) or CRISPR-Cas9 gene editing (sg-Con vs sg-1# and sg-2#) in Pool2 or HR20 cells. The expression of IRS1, p-IGF-1R, IGF-1R, p-Akt(T308), p-Akt(S473), Akt, p-S6K, S6K, p-FOXO3a, FOXO3a, and β-actin was measured by western blot assays. **f** SKBR3 and BT474 cells were stably transfected with control vector (pLEX-Con or Con) or IRS1-overexpressing vector (pLEX-IRS1 or IRS1) followed by treatment with rhIGF2 (80 ng/ml) in combination with Herceptin at indicated concentrations for 72 h. Cell viability was evaluated by MTS assays, SKBR3 IRS1 + IGF2: **$p = 0.0059$, ***$p = 0.0002$, ***$p = 0.0003$, BT474 IRS1 + IGF2: **$p = 0.0038$, ***$p = 0.0004$. $n = 3$ biological independent samples (**b–d**, **f**). Data are presented as mean values ± SEM (**b–d**, **f**). Statistical significance was determined by a two-tailed Student's *t*-test (**b–d**, **f**). Data show a representative of three independent experiments (**a**, **e**). All data are provided in the Source Data file.

**FOXO3a regulation of IRS1 dictates a negative feedback inhibition of IGF-1R signaling**. It has been shown that a negative feedback inhibition loop is critical for IGF-induced signaling in cellular homeostasis[28]. To investigate whether a similar mechanism exists in HER2-positive breast cancer cells to control IGF-1R signaling, we treated SKBR3 and BT474 cells with a series of doses of rhIGF2. At 5–20 ng/ml, rhIGF2 induced a dose-dependent activation of IGF-1R/Akt/mTOR signaling along with increase of IRS1 protein, but not mRNA levels in SKBR3 cells. However, high doses (40–80 ng/ml) of rhIGF2 attenuated activities of Akt and mTOR kinases (p-IGF-1R levels remained unchanged) and decreased IRS1 protein, but not mRNA levels (Fig. 2a and Supplementary Fig. 2a). Similar results i.e., rhIGF2-induced biphasic effects on IGF-1R/Akt/mTOR signaling and IRS1 expression were also observed in BT474 cells (Supplementary Fig. 2b). These data suggested that IGF2 at high dose (80 ng/ml) elicited a negative feedback inhibition of the IGF-1R/Akt/mTOR signaling. To assess the role of IRS1 in this feedback regulation, we transfected an *IRS1* cDNA expression vector into SKBR3 and BT474 cells. Ectopic expression of IRS1 did not alter the effect of rhIGF2 on p-IGF-1R levels, but dramatically rescued rhIGF2 (80 ng/ml)-induced reduction of p-Akt[(T308)], p-Akt[(S473)], p-S6K, and p-FOXO3a in both SKBR3 (Fig. 2b) and BT474 cells (Supplementary Fig. 2c). Taken together, our data indicated that IRS1 expression was required for maintaining rhIGF2 (80 ng/ml)-induced negative feedback inhibition of the IGF-1R/Akt/mTOR signaling.

Next, we wondered whether FOXO3a, a key transcription factor downstream target of Akt/mTOR signaling, might be involved in IGF2-induced feedback regulation of IRS1. Knockdown of FOXO3a expression via specific shRNAs not only increased the basal levels of IRS1, it also abolished rhIGF2 (80 ng/ml)-induced downregulation of IRS1 in both SKBR3 (Fig. 2c) and BT474 cells (Supplementary Fig. 2d). In addition, *FOXO3a* gene deletion by CRISPR-Cas9 technology had similar effects as specific knockdown of FOXO3a by shRNAs on IRS1 expression (Fig. 2d and Supplementary Fig. 2e). These results supported the notion that IRS1 expression was driven by FOXO3a, which might be an important mechanism controlling IGF2 (80 ng/ml)-induced negative feedback inhibition of the signaling in HER2-positive breast cancer cells. Interestingly, specific knockdown of FOXO3a by shRNAs or *FOXO3a* gene deletion either alone or in combination with rhIGF2 (high and low doses) treatment had little effect on *IRS1* mRNA expression (Supplementary Fig. 2f). One plausible hypothesis would be that FOXO3a modulated specific miRNAs targeting *IRS1* to influence its protein translation.

**FOXO3a regulates IRS1 expression via specific miRNAs**. We then sought to identify *IRS1*-targeting miRNAs potentially regulated by FOXO3a. Bioinformatics analysis (http://www.targetscan.org) predicted a number of miRNAs with conserved binding sites in the 3′-UTR of *IRS1* mRNA. We were particularly interested in miR-128-3p and miR-30a-5p (Supplementary Fig. 3a), because low doses (5–20 ng/ml) of rhIGF2 downregulated, whereas rhIGF2 at high dose (80 ng/ml) upregulated the miRNAs in SKBR3 and BT474 cells (Supplementary Fig. 3b). However, the expression levels of miR-191-5p, which is not predicted to target *IRS1*, remained unchanged upon rhIGF2 treatments (Supplementary Fig. 3b). To determine whether the alterations of miR-128-3p and miR-30a-5p played a role in rhIGF2 regulation of IRS1, we took advantage of specific inhibitors and mimics of miR-128-3p and miR-30a-5p. Combinations of the inhibitors of miR-128-3p and miR-30a-5p potently increased IRS1, whereas overexpression of the miRNAs via mimic transfection markedly decreased IRS1 in both SKBR3 and BT474 cells (Fig. 3a). Importantly, while one

miRNA inhibitor partially, combinations of two miRNA inhibitors ultimately rescued rhIGF2 (80 ng/ml)-induced downregulation of IRS1 (Fig. 3b). It seemed that low dose of rhIGF2 (10 ng/ml) profoundly downregulated miR-128-3p and miR-30a-5p to such a low level (Fig. 3c), so that the miRNA inhibitors no longer influenced rhIGF2 (10 ng/ml)-induced upregulation of IRS1 (Fig. 3b). To study whether miR-128-3p and miR-30a-5p directly targeted *IRS1*, we cloned the *IRS1* mRNA 3′-UTR containing either wild type or mutant binding site of miR-128-3p or miR-30a-5p into the luciferase reporter vector. The combinations of two miRNA inhibitors as compared to one miRNA inhibitor more potently increased luciferase activity of the reporter with wild type binding site, but not the one with mutant binding site (Supplementary Fig. 3c). Additionally, HEK293T cells were co-transfected with the luciferase reporter and mimics of miR-128-3p and/or miR-30a-5p. While single miRNA mimic attenuated, combinations of two miRNA mimics profoundly suppressed luciferase activity of the *IRS1* reporter with wild type, but not mutant binding site (Supplementary Fig. 3d).

Next, we investigated how FOXO3a regulated expression of miR-128-3p and miR-30a-5p. Downregulation of FOXO3a by either specific shRNAs or CRISPR-Cas9 gene editing not only reduced basal levels of the miRNAs, but also abrogated rhIGF2 (80 ng/ml)-induced upregulation of the miRNAs (Fig. 3c and Supplementary Fig. 3e), suggesting that rhIGF2 regulated expression of miR-128-3p and miR-30a-5p dependent upon FOXO3a. By careful examination of the miRNAs' promotors, we noticed that there are three FOXO3a-binding sites (A, B, and C) within 2 kb region upstream of the precursor start sites for both miR-128-3p and miR-30a-5p (Fig. 3d). We then studied whether FOXO3a regulated the miRNA expression via the predicted binding sites. Chromatin immunoprecipitation (ChIP) followed by qPCR assays revealed that FOXO3a was markedly enriched at sites A and B in miR-128-3p promoter, and at sites A and C in miR-30a-5p promoter (Fig. 3e). In addition, low dose (10 ng/ml) of rhIGF2 significantly decreased, whereas high dose (80 ng/ml) of rhIGF2 dramatically increased FOXO3a enrichment at the miRNA promoters (Fig. 3e). For further confirmation, we constructed pGL4 luciferase reporters containing the miRNAs' promoters with wild type or mutant FOXO3a-binding sites. We found that low dose (10 ng/ml) of rhIGF2 repressed, whereas high dose (80 ng/ml) of rhIGF2 enhanced the luciferase activity of wild type reporter in SKBR3 cells (Supplementary Fig. 3f). In contrast, rhIGF2 treatment had no effect on the luciferase activity of mutant reporter. Moreover, specific knockdown of FOXO3a reduced luciferase activity and completely abrogated the increase of luciferase activity-induced by high dose (80 ng/ml) of rhIGF2 (Supplementary Fig. 3f). Taken together, these data demonstrated a crucial role for miR-128-3p and miR-30a-5p transcriptionally regulated by FOXO3a in control of IRS1 expression in HER2-positive breast cancer cells.

**PPP3CB-mediated reduction of p-FOXO3a alters the feedback inhibition**. Akt[(473)]/mTORC2-mediated phosphorylation of FOXO3a results in inactivation of its transcriptional activity[29]. Next, we sought to explore the underlying mechanism that altered p-FOXO3a levels during IGF2-trggered negative feedback inhibition of IGF-1R/Akt/mTOR signaling. We first studied the potential involvement of mTOR with a specific inhibitor WAY-600, which effectively inhibits both mTOR1 and mTOR2 activity[30]. WAY-600 disrupted rhIGF2-induced changes of p-S6K, p-Akt[(473)], and p-FOXO3a levels in both SKBR3 and BT474 cells. It also reduced the basal levels of IRS1 and abolished IRS1 upregulation-induced by low dose (10 ng/ml) of rhIGF2 (Supplementary Fig. 4a). WAY-600 potently induced expression of miR-128-3p and miR-30a-5p,

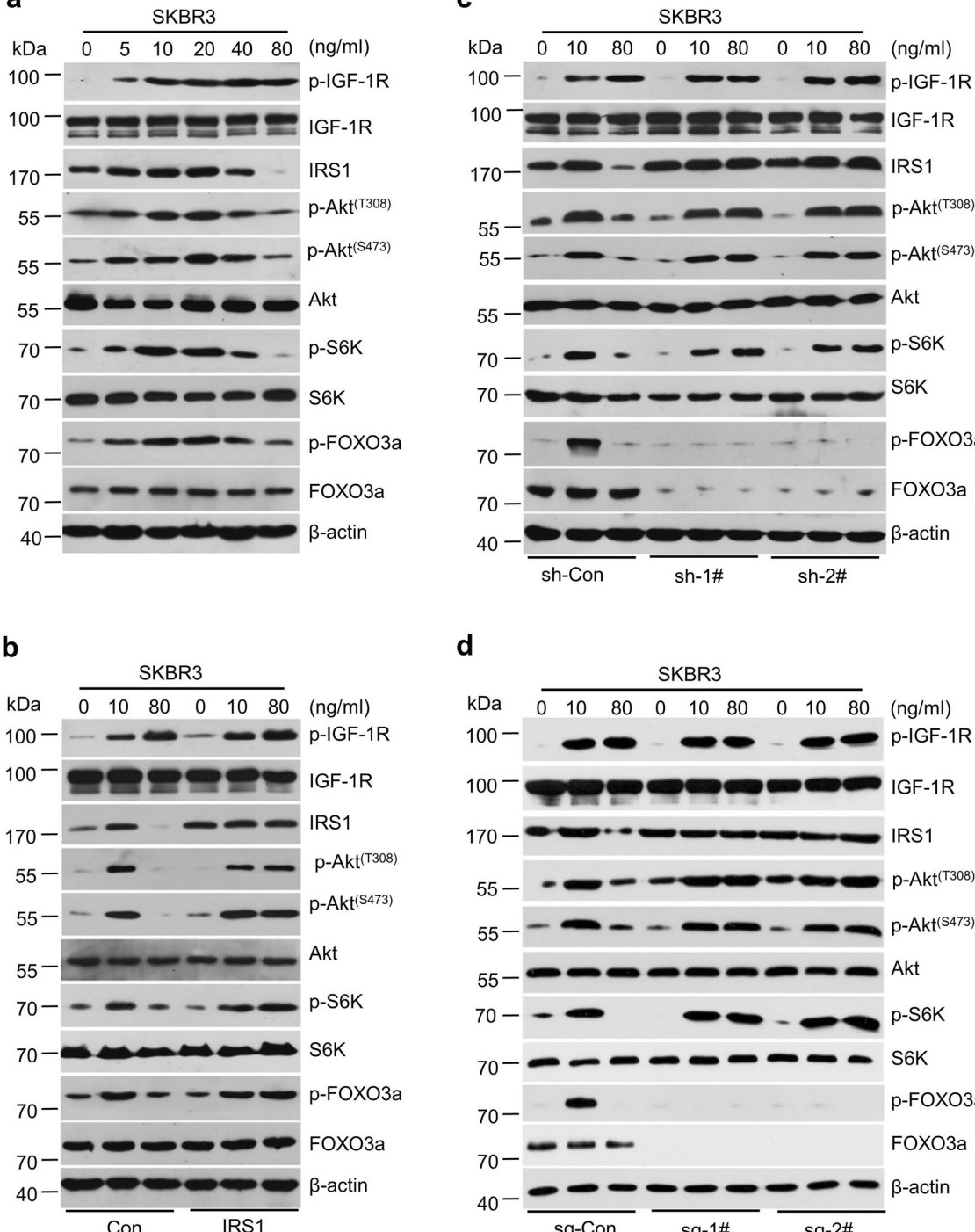

**Fig. 2 FOXO3a and IRS1 regulated a negative feedback inhibition of IGF2-triggered IGF-1R/Akt/mTOR signaling in HER2-positive breast cancer cells.** **a** SKBR3 cells were treated with rhIGF2 at indicated concentrations for 6 h. The expression of p-IGF-1R, IGF-IR, p-Akt(S473), p-Akt(T308), Akt, p-S6K, S6K, p-FOXO3a, FOXO3a, and β-actin was examined by western blot assays. **b** SKBR3 cells were transfected with a control empty vector (Con) or the same vector containing an *IRS1* cDNA (IRS1), followed by rhIGF2 treatment for 6 h. The expression of indicated proteins was examined by western blot assays. **c** SKBR3 cells were transfected with control shRNA (sh-Con) or specific *FOXO3a* shRNAs (sh-1# and sh-2#) followed by rhIGF2 treatment for 6 h. The expression of indicated proteins was examined by western blot assays. **d** SKBR3 cells with *IRS1* gene deletion via CRISPR-Cas9 (sg-Con vs sg-1# and sg-2#) were treated by rhIGF2 treatment for 6 h. The expression of indicated proteins were examined by western blot assays. Data show a representative of three independent experiments (**a**–**d**). All data are provided in the Source Data file.

but not miR-191-5p and profoundly attenuated the response of SKBR3 and BT474 cells to rhIGF2 (Supplementary Fig. 4b). We then performed additional assays with a more selective inhibitor against mTOR1 - rapamycin[29], which persistently inhibited mTOR1 activity evidenced by the decrease of p-S6K in the absence

and presence of low dose (10 ng/ml) of rhIGF2 (Fig. 4a). More importantly, rapamycin completely abrogated high dose (80 ng/ml) of rhIGF2-induced reduction of p-FOXO3a (Fig. 4a), and switched the expression of miR-128-3p and miR-30a-5p from upregulation to downregulation in SKBR3 and BT474 cells (Fig. 4b). Treatments

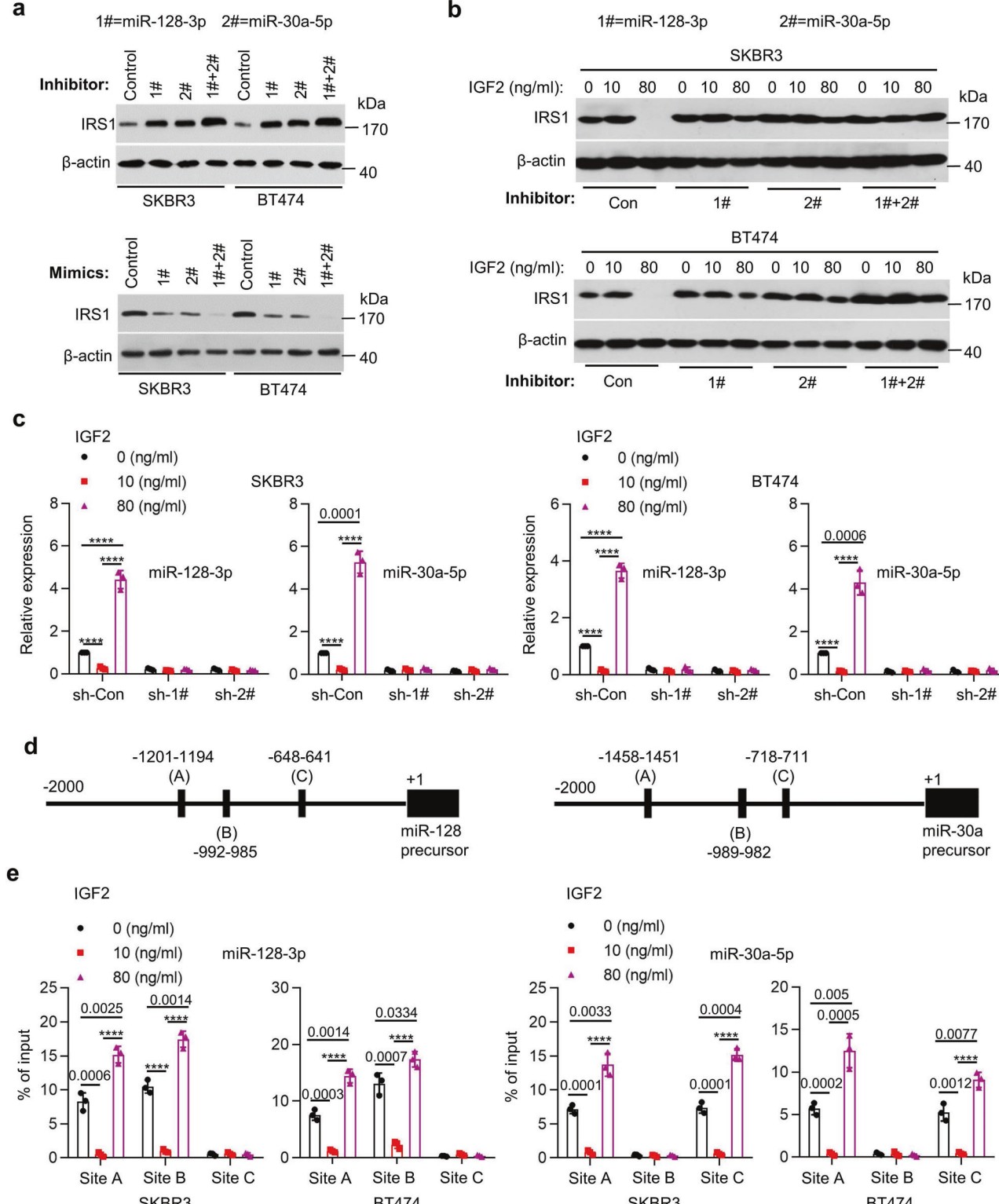

of SKBR3 and BT474 cells with rapamycin either alone or in combination with rhIGF2 (low or high dose) had no effect on the expression of miR-191-5p (Supplementary Fig. 4c).

We wondered if our observed alterations of p-FOXO3a might be attributed to some phosphatases. To this end, we took advantage of a series of phosphatase inhibitors with an aim to identify the phosphatase(s) altering FOXO3a's transcriptional activity (phosphorylation). Treatment of SKBR3 and BT474 cells with Cypermethrin and Deltamethrin, two specific inhibitors of

the serine/threonine-protein phosphatase 2B (PP2B)[31], reversed the decrease of p-FOXO3a in response to rhIGF2 (80 ng/ml) (Fig. 4c). PP2B, also known as calcineurin (CN), is composed of a catalytic subunit (CNA) and a regulatory subunit (CNB)[32]. CNA has three isoforms: CNAα (PPP3CA), CNAβ (PPP3CB) and CNAγ (PPP3CC), whereas CNB has two isoforms: CNBα (PPP3R1) and CNBβ (PPP3R2)[31]. We discovered that rhIGF2 treatment of SKBR3 and BT474 cells induced expression of PPP3CB, but not other isoforms, in a dose-dependent manner

**Fig. 3 FOXO3a controlled miRNAs to influence IRS1 expression and the negative feedback suppression of the IGF-1R/Akt/mTOR signaling. a** SKBR3 or BT474 cells were transfected with miR-128-3p or/and miR-30a-5p inhibitor (upper) or transfected with miR-128-3p or/and miR-30a-5p mimics (bottom). The expression of IRS1 and β-actin was examined by western blot assays. **b** SKBR3 or BT474 cells were transfected with miR-128-3p or/and miR-30a-5p inhibitor followed by rhIGF-2 treatment at indicated concentration for 24 h. The expression of IRS1 and β-actin was examined by western blot assays. **c** SKBR3 or BT474 cells were transfected with FOXO3a shRNA followed by rhIGF-2 treatment at indicated concentration for 24 h. The expression levels of miR-128-3p and miR-30a-5p were measured by qRT-PCR, ****$p < 0.0001$. **d** A schematic representation of FOXO3a binding sites within the 2 kb putative promoters of miR-128-3p and miR-30a-5p. The first base of the precursors of miR-128-3p and miR-30a-5p is defined as '+1'. **e** SKBR3 or BT474 cells were treated with rhIGF-2 at indicated concentrations for 24 h. The enrichment of FOXO3a at miR-128 or miR-30a promoter was evaluated by ChIP-qPCR. The chromatin was precipitated with an anti-FOXO3a antibody. The precipitated chromatin was then analyzed by qRT-PCR with primers specific for the putative FOXO3a binding sites, ****$p < 0.0001$. $n = 3$ biological independent samples (**c**, **e**). Data are presented as mean values ± SEM (**c**, **e**). Statistical significance was determined by a two-tailed Student's $t$-test (**c**, **e**). Data show a representative of three independent experiments (**a**, **b**). All data are provided in the Source Data file.

(Fig. 4d and Supplementary Fig. 4d). It had no effect on *PPP3CB* mRNA (Supplementary Fig. 4e). Moreover, rapamycin abolished high dose of rhIGF2-induced upregulation of PPP3CB (Fig. 4a). As expected, specific knockdown of PPP3CB significantly decreased the enrichment of FOXO3a induced by high dose (80 ng/ml) of rhIGF2 at the promoters of miR-128-3p and miR-30a-5p (Fig. 4e, f). Collectively, these data suggested that PPP3CB likely regulated p-FOXO3a levels to alter its transcriptional activity, which in turn influenced the negative feedback inhibition of IGF2-trggered IGF-1R/Akt/mTOR signaling.

**Dysregulation of FOXO3a-miRNA axis leads to Herceptin resistance.** The *IRS1*-targeting miR-128-3p and miR-30a-5p played an important role in restricting IRS1 expression in SKBR3 and BT474 cells (Fig. 3), and we observed a striking increase of IRS1 in pool2 and HR20 cells (Fig. 1a). Thus, it was conceivable to hypothesize that the IRS1 upregulation might be due to dysregulation of miR-128-3p and miR-30a-5p in Herceptin-resistant cells. To test this hypothesis, we examined the miRNAs' expression in Herceptin-sensitive and -resistant cells. As expected, miR-128-3p and miR-30a-5p were significantly downregulated in pool2 and HR20 cells than that in SKBR3 and BT474 cells, respectively (Fig. 5a). While one miRNA mimic attenuated, combinations of two-miRNA mimics potently diminished IRS1, which accompanied with p-FOXO3a reduction (Fig. 5b). Furthermore, over-expression of the miRNAs significantly re-sensitized pool2 cells to Herceptin-mediated growth inhibition, which was abrogated by ectopic expression of IRS1 (Fig. 5c), suggesting that the miRNAs-mediated Herceptin sensitization mainly through their inhibitory effects on *IRS1*. However, inhibition of miR-128-3p and/or miR-30a-5p alone had no effect on Herceptin sensitivity, whereas the miRNA inhibitors in combination with high dose (80 ng/ml) of rhIGF2 successfully converted SKBR3 into Herceptin-resistant cells. Moreover, specific knockdown of IRS1 abolished the resistance phenotype-induced by simultaneous treatment with the miRNA inhibitors and rhIGF2 (Fig. 5c). Similar results were also observed in HR20 and BT474 cells (Supplementary Fig. 5a). Thus, our data indicated that IRS1 and IGF2 were functionally interdependent to control Herceptin sensitivity in HER2-positive breast cancer cells.

Meanwhile, we noticed that IGF2 levels were significantly increased in the CM of Herceptin-resistant cells (Fig. 1b). To determine whether the upregulation of IGF2 was also mediated by specific miRNAs, we performed target scan (http://www.targetscan.org) analysis to identify the miRNAs that have conserved binding sites in the 3′-UTR of *IGF2* mRNA. The expression of miR-193a-5p was significantly reduced in pool2 and HR20 cells (Fig. 5d). Overexpression of miR-193a-5p profoundly decreased IGF2 levels in the CM of resistant cells. In contrast, the specific inhibitor of miR-193-5p significantly increased IGF2 levels in the CM of sensitive cells (Fig. 5e). To study if miR-193a-5p

directly targeted *IGF2*, we cloned the 3′-UTR of *IGF2* mRNA containing either wild type or mutant miR-193a-5p binding site into the pMIR-REPORT vector. We found that increased miR-193a-5p via mimic transfection repressed the luciferase activity of the reporter with wild type, but not mutant binding site in pool2 and HR20 cells (Supplementary Fig. 5b). Similar results were also obtained from HEK293T cells transfected with miR-193-5p mimic. The specific inhibitor of miR-193-5p enhanced luciferase activity of the reporter with wild type, but not mutant binding site in SKBR3 and BT474 cells (Supplementary Fig. 5b).

Given that FOXO3a regulated expression of miR-128-3p and miR-30a-5p in SKBR3 and BT474 cells (Supplementary Fig. 3e and Fig. 4) and that both mTOR1 and mTOR2 were activated in Herceptin-resistant cells (Fig. 1a), it was necessary to determine if mTOR2-mediated inactivation of FOXO3a led to down-regulation of miR-128-3p, miR-30a-5p, and miR-193a-5p in the resistant cells. Treatment of pool2 and HR20 cells with WAY-600, which inhibited both mTOR1 and mTOR2, significantly increased expression of miR-128-3p, miR-30a-5p, and miR-193a-5p (Fig. 5f). However, a more selective mTOR1 inhibitor, rapamycin had no effect on the miRNAs' expression (Supplementary Fig. 5c). Interestingly, specific knockdown of FOXO3a blocked the miRNAs' increase in response to WAY-600 (Supplementary Fig. 5d), suggesting that dysregulation of the miRNAs in Herceptin-resistant cells might be attributed to FOXO3a inactivation. FOXO3a has two conserved binding sites (A and B) within the 2 kb region upstream of miR-193a-3p precursor start site (Supplementary Fig. 5e). Studies with ChIP-qPCR revealed that FOXO3a was considerably enriched at site A in SKBR3 and BT474 cells, but not that in pool2 and HR20 cells (Supplementary Fig. 5f). However, WAY-600 significantly enhanced FOXO3a enrichment at the promoter of miR-193a-5p in pool2 and HR20 cells (Fig. 5g). We then constructed pGL4 luciferase reporters containing miR-193a-5p promoter with wild type or mutated FOXO3a binding sites. The luciferase activity of wild type reporter in pool2 and HR20 cells was much lower than that in SKBR3 and BT474 cells, respectively (Supplementary Fig. 5g). WAY-600 increased the luciferase activity-driven by miR-193a-5p promoter with wild type, but not mutant FOXO3a binding sites (Supplementary Fig. 5h). Collectively, these data established an essential role for FOXO3a transcriptional regulation of specific miRNAs to control IGF2 and IRS1 expression, thereby altering Herceptin sensitivity in HER2-positive breast cancer cells.

**The p-STAT6/HDAC1 complex represses *PPP3CB* in Herceptin-resistant breast cancer.** Our studies suggested that the expression of PPP3CB in SKBR3 and BT474 cells critically regulated FOXO3a-miRNAs axis to control the negative feedback inhibition of IGF2-trggered IGF-1R signaling. It was unclear, however, whether PPP3CB played any role in Herceptin resistance. We discovered a

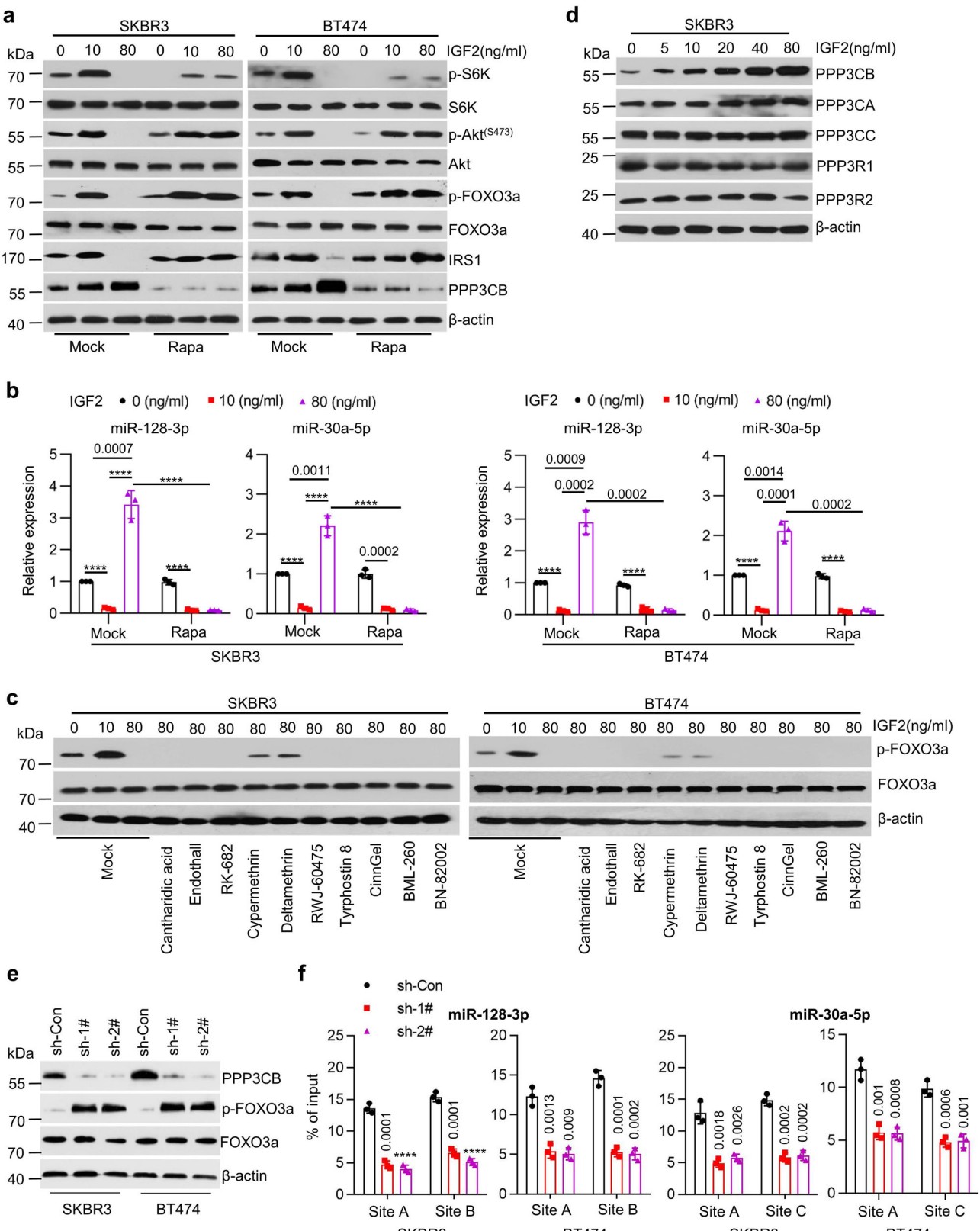

significant downregulation of PPP3CB at both mRNA and protein levels in Herceptin-resistant cells (Fig. 6a). Further studies revealed that specific knockdown of PPP3CB expression resulted in resistance to Herceptin treatment of SKBR3 and BT474 cells (Fig. 6b). In contrast, overexpression of PPP3CB not only re-sensitized the resistant cells to Herceptin, but also enhanced the efficacy of

Herceptin against the parental cells (Fig. 6b and Supplementary Fig. 6a). In addition, ectopic expression of PPP3CB dramatically decreased p-FOXO3a and IRS1, and markedly reduced IGF2 levels in the CM of Herceptin-resistant cells (Fig. 6c). Meanwhile, over-expression of PPP3CB significantly upregulated miR-128-3p, miR-30a-5p, and miR-193a-5p in the resistant cells (Fig. 6d). These

**Fig. 4 PPP3CB-mediated decrease of p-FOXO3a levels altered the feedback regulation. a, b** SKBR3 or BT474 cells were treated with vehicle (Mock) or Rapamycin (10 μM, Rapa) in combination with rhIGF2 at indicated concentrations for 24 h. The expression of p-Akt$^{(S473)}$, Akt, p-S6K, S6K, p-FOXO3a, FOXO3a, IRS1, PPP3CB, and β-actin was examined by western blot assays (**a**); The expression levels of miR-128-3p and miR-30a-5p were detected by qRT-PCR (**b**), ****$p < 0.0001$. **c** SKBR3 or BT474 cells were treated with a series of phosphatase inhibitors as Cantharidic acid (0.5 μM), Endothall (1 μM), RK-682 (10 μM), Cypermethrin (1 μM), Deltamethrin (1 μM), RWJ-60475 (2 μM), Tyrphostin 8 (10 μM), CinnGel (1 μM), BML-260 (10 μM), or BN-82002 (5 μM) in combination with rhIGF2 at indicated concentrations for 24 h. The expression of p-FOXO3a, FOXO3a, and β-actin was analyzed by western blot assays. **d** SKBR3 cells were treated with rhIGF2 at indicated concentrations for 24 h. The expression of PPP3CB, PPP3CA, PPP3CC, PPP3R1, PPP3R2, and β-actin was examined by western blot assays. **e, f** SKBR3 or BT474 cells with specific knockdown of PPP3CB by shRNAs (sh-Con vs sh-1# and sh-2#) were treated with rhIGF2 (80 ng/ml) for 24 h. The expression of PPP3CB, p-FOXO3a, FOXO3a and β-actin were examined by western blot assays (**e**); The enrichment of FOXO3a at the promoters of miR-128-3p and miR-30a-5p were detected by ChIP-qPCR (**f**), ****$p < 0.0001$. $n = 3$ biological independent samples (**b, f**). Data are presented as mean values ± SEM (**b, f**). Statistical significance was determined by a two-tailed Student's $t$-test (**b, f**). Data show a representative of three independent experiments (**a, c, d, e**). All data are provided in the Source Data.

data suggested that downregulation of PPP3CB in Herceptin-resistant cells might play an important role leading to dysregulation of FOXO3a-miRNA axis. To explore the molecular basis of PPP3CB downregulation, we first considered an epigenetic mechanism as we showed that a class I histone deacetylase (HDAC) inhibitor, entinostat (SNDX-275 or MS-275) mainly inhibited HDAC1 and potently induced apoptosis in HER2-positive breast cancer cells[33]. We found that entinostat treatment of pool2 and HR20 cells increased the protein (Fig. 6e) and mRNA expression (Supplementary Fig. 6b) of PPP3CB. It also reduced the levels of p-FOXO3a and IRS1 in the cells (Fig. 6e) and IGF2 in the CM (Supplementary Fig. 6c). Consistently, entinostat upregulated miR-128-3p, miR-30a-5p, and miR-193a-5p in pool2 and HR20 cells (Supplementary Fig. 6d). We then analyzed the promoter of PPP3CB to identify the potential factor(s) that might coordinate with HDAC1 to control PPP3CB transcription. We detected two conserved binding sites for signal transducer and activator of transcription 6 (STAT6) (Supplementary Fig. 6e). While there was no difference of total STAT6 between Herceptin-resistant and -sensitive cells, p-STAT6 levels were clearly increased in the resistant cells (Fig. 6a). To determine the importance of p-STAT6 in regulation of PPP3CB expression, we used specific shRNAs to inhibit STAT6 expression (Supplementary Fig. 6f). Specific knockdown of STAT6 markedly enhanced protein and mRNA expression of PPP3CB (Fig. 6f and Supplementary Fig. 6g) in Herceptin-resistant cells. It also downregulated p-FOXO3a and IRS1 (Fig. 6f), reduced IGF2 levels in the CM (Supplementary Fig. 6h) of the resistant cells, and upregulated miR-128-3p, miR-30a-5p, and miR-193a-5p (Supplementary Fig. 6i). Moreover, ChIP-qPCR analysis revealed that p-STAT6 was enriched at its predicted binding sites (A and B) in PPP3CB promoter in Herceptin-resistant, but not -sensitive cells (Supplementary Fig. 6j). Specific knockdown of STAT6 significantly reduced the recruitment of HDAC1 to PPP3CB promoter in the resistant cells (Fig. 6g). In addition, Co-IP assays confirmed the interaction between p-STAT6 and HDAC1 in Herceptin-resistant, but not -sensitive cells (Fig. 6h). Collectively, our data demonstrated that p-STAT6 interacted with HDAC1 to repress PPP3CB gene transcription in Herceptin-resistant breast cancer cells.

Next, we sought to explore the mechanism responsible for the increase of p-STAT6 in Herceptin-resistant cells. In agreement with our previous report[24], we confirmed that p-Src were increased in the resistant cells (Fig. 6a). We found that treatment with the Src inhibitor SU6656 decreased p-STAT6 and increased both mRNA (Supplementary Fig. 6k) and protein levels of PPP3CB, accompanied with reduction of p-FOXO3a and IRS1 in the resistant cells (Fig. 6i) and reduced IGF2 levels in the CM (Supplementary Fig. 6l). SU6656 treatment also upregulated miR-128-3p, miR-30a-5p, and miR-193a-5p (Supplementary Fig. 6m), and significantly reduced the enrichment of p-STAT6 and HDAC1 at PPP3CB promoter (Supplementary Fig. 6n). Taken together, our data supported the notion that activation of Src

kinase (evidenced by increased p-Src) likely promoted STAT6 phosphorylation to enhance its interaction with HDAC1, and subsequently the p-STAT6/HDAC1 complex acted in concert to suppress PPP3CB expression in Herceptin-resistant breast cancer cells.

**PPP3CB-FOXO3a axis controls Herceptin efficacy in vivo**. We next examined the role of IRS1 in determining the efficacy of Herceptin against HER2-positive breast cancer in vivo. Pool2 cells stably transfected with control shRNA or IRS1-specific shRNA were subcutaneously inoculated into nude mice to establish tumor xenografts. When tumor volumes reached ~100 mm³, the mice were randomly grouped and received IP (intraperitoneal) injection of either Herceptin (10 mg/kg) or PBS (0.2 ml) every 5 days. Specific knockdown of IRS1 alone had no effect on tumor growth as compared to the controls. Herceptin alone exhibited an initial inhibition on tumor growth. The tumors then re-grew at a similar rate as the controls, indicating that pool2 cells retained their Herceptin resistance phenotype in vivo. Importantly, Herceptin in combination with specific knockdown of IRS1 led to a sustained and significant inhibition on tumor growth and dramatically reduced tumor sizes and weights (Fig. 7a and Supplementary Fig. 7a). Similar results were also obtained with IRS1 gene deletion by CRISPR-Cas9 technology, i.e. IRS1 deletion reversed the resistance phenotype and potently enhanced Herceptin-mediated anti-tumor activity in vivo (Fig. 7b and Supplementary Fig. 7b). Additionally, we found that specific knockdown of IRS1 expression or IRS1 gene deletion decreased the levels of p-Akt and p-FOXO3a in vivo (Supplementary Fig. 7c), confirming that p-Akt/p-FOXO3a acted as downstream of IRS1 in Herceptin-resistant breast cancer cells. To determine the clinical significance of our findings, we then performed analyses with serum and tumors obtained from 23 HER2-positive breast cancer patients who responded well to Herceptin-containing treatments and 17 HER2-positive breast cancer patients poorly responded to the treatments. IGF2 levels in the serum of patients with a poor response were significantly higher than that of the patients with a good response (Fig. 7c). Immunohistochemistry (IHC) assays revealed that the tumors obtained from patients with a poor response to Herceptin-containing treatments had much higher levels of IRS1 and p-FOXO3a and significantly lower expression of PPP3CB (Fig. 7d and Supplementary Fig. 7d). The p-FOXO3a levels were positively correlated with IRS1 expression, whereas negatively correlated with PPP3CB expression in all tumors we tested (Supplementary Fig. 7e). Moreover, bioinformatics analysis of the databases of GEO, EGA, and TCGA using the Kaplan–Meier plotter (https://kmplot.com/analysis/index.php?p=service&cancer=breast) showed that low expression of PPP3CB, as compared to high PPP3CB expression, significantly associated with a worse overall survival (OS), relapse-free survival (RFS), and distant metastasis-free survival (DMFS) in breast cancer patients (Fig. 7e). These data support that PPP3CB likely functioned as a

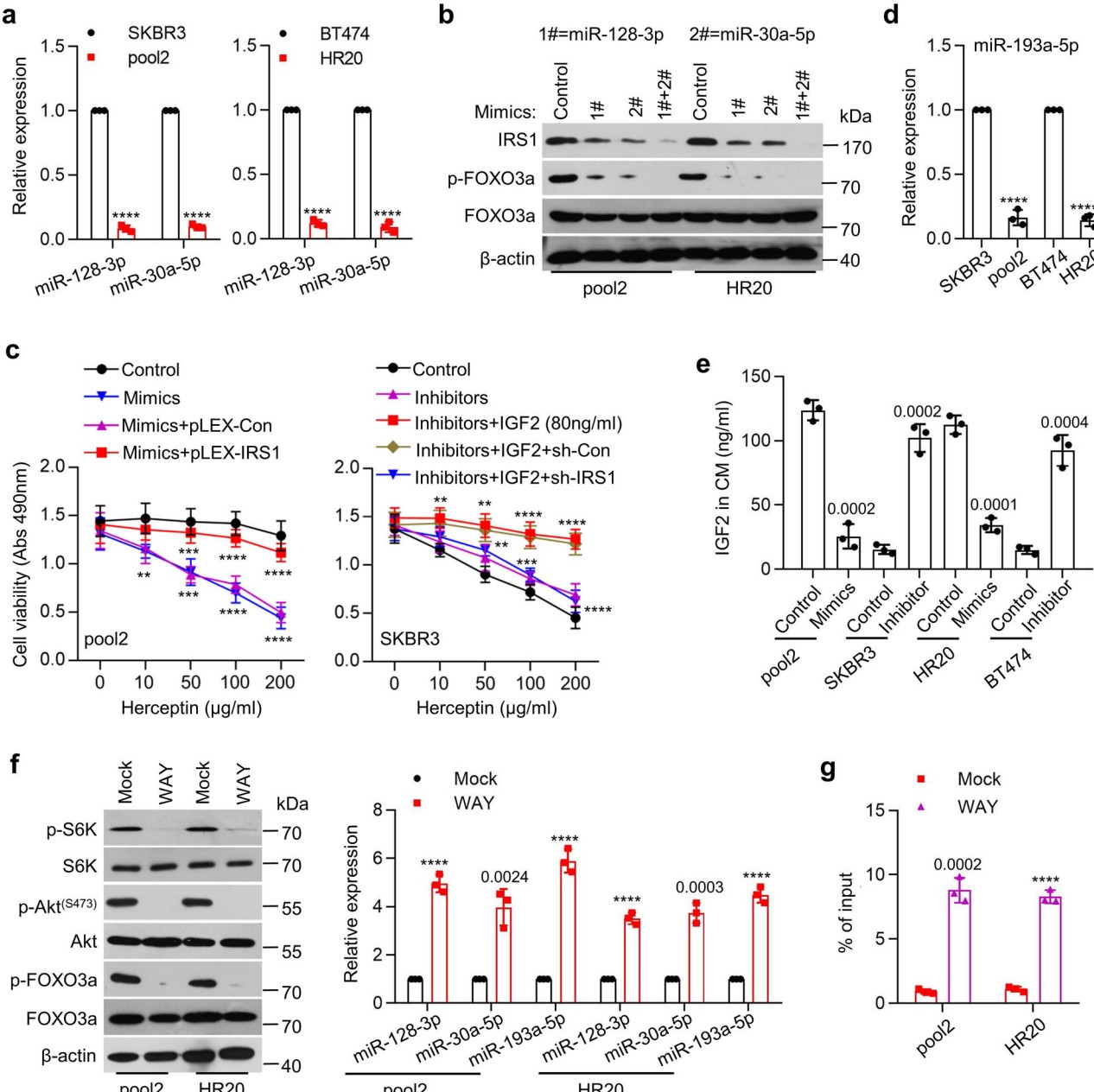

**Fig. 5 Dysregulation of FOXO3a-miRNAs axis conferred Herceptin resistance. a** The expression levels of miR-128-3p and miR-30a-5p in SKBR3, pool2, BT474, or HR20 cells were measured by qRT-PCR, ****$p < 0.0001$. **b** Pool2 or HR20 cells were transfected with miR-128-3p or/and miR-30a-5p mimics. The expression of IRS1, p-FOXO3a, FOXO3a, and β-actin was examined by western blot assays. **c** Pool2 cells were transfected with miRNA mimics in combination with pLEX-IRS1 followed by Herceptin treatment at indicated concentrations for 72 h (left). SKBR3 cells were transfected miRNA inhibitors in combination with IRS1 shRNA, and then treated with 80 ng/ml rhIGF2 along with Herceptin (right). Cell viability was evaluated by MTS assays. Pool2 Mimics: **$p = 0.0022$, ***$p = 0.0003$, ****$p < 0.0001$, Mimics+pLEX-IRS1: ***$p = 0.0001$, ****$p < 0.0001$, SKBR3 Inhibitors+IGF2: **$p = 0.0072$, **$p = 0.0035$, ****$p < 0.0001$, Inhibitors+IGF2 + sh-IRS1: **$p = 0.0079$, ***$p = 0.0002$, ****$p < 0.0001$. **d** The expression levels of miR-193a-5p in SKBR3, pool2, BT474, and HR20 cells were detected by qRT-PCR, ****$p < 0.0001$. **e** Pool2 or HR20 cells were transfected with a miR-193a-5p mimic. SKBR3 or BT474 were transfected with a miR-193a-5p inhibitor. IGF2 levels in the CM were measured by ELISA. **f** Pool2 or HR20 cells were treated with vehicle (Mock) or WAY-600 (1 μM, WAY). The expression of p-Akt(S473), Akt, p-S6K, S6K, p-FOXO3a, FOXO3a, and β-actin was examined by western blot assays (left); the expression levels of miR-128-3p, miR-30a-5p and miR-193a-5p were detected by qRT-PCR (right), ****$p < 0.0001$. **g** Pool2 or HR20 cells were treated with WAY-600 for 24 h. The enrichment of FOXO3a at miR-193a-5p promoter was examined by ChIP-qPCR assays, ****$p < 0.0001$. $n = 3$ biological independent samples (**a**, **c–e**, **g**). Data are presented as mean values ± SEM (**a**, **c–e**, **g**). Statistical significance was determined by a two-tailed Student's t-test (**a**, **c–e**, **g**). Data show a representative of three independent experiments (**b**, **f**). All data are provided in the Source Data.

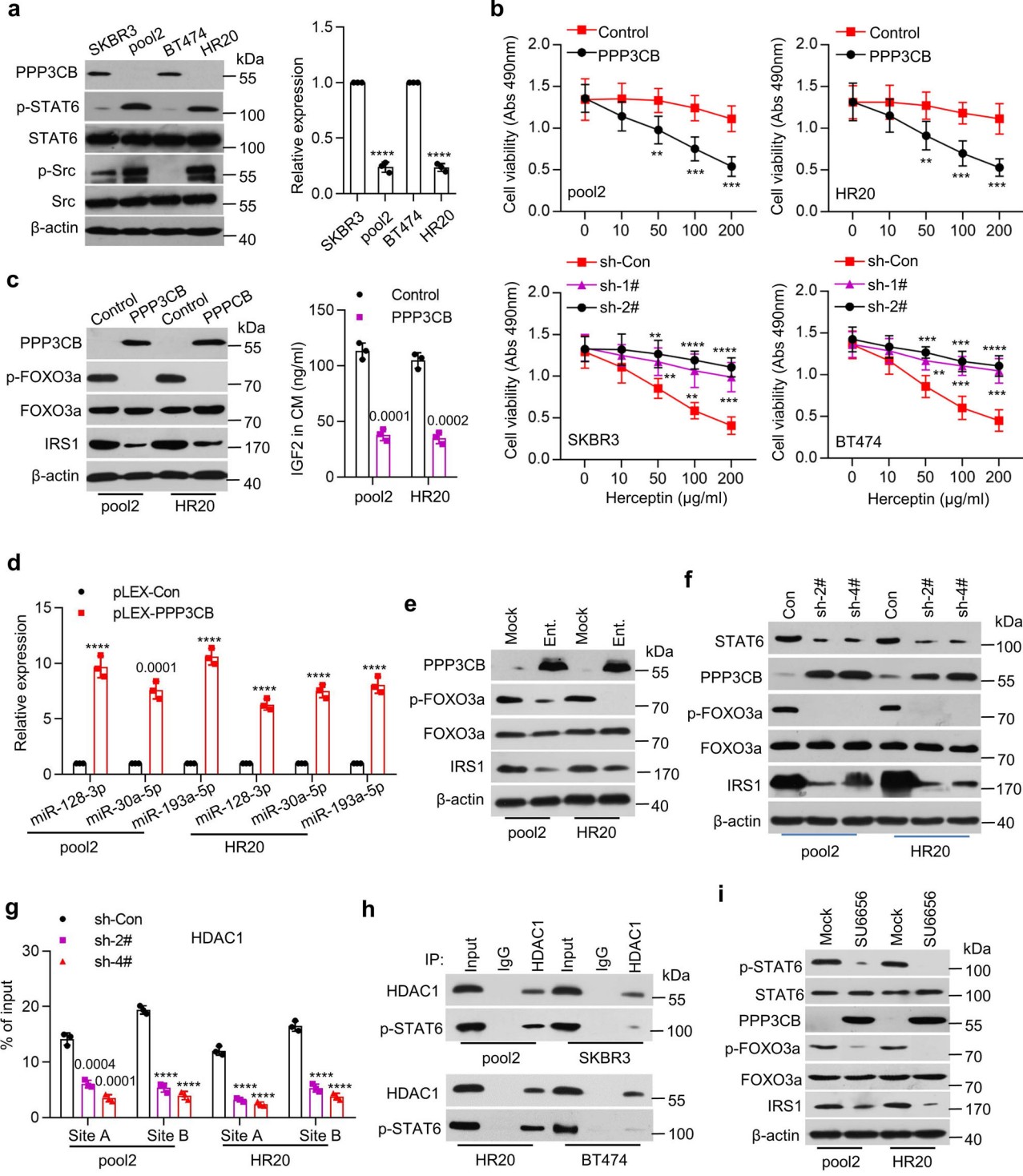

tumor suppressor in breast cancer progression. Collectively, our studies demonstrated that dysregulation of PPP3CB-FOXO3a axis resulted in increased expression of both IGF2 and IRS1, which in turn disrupted the negative feedback inhibition loop of IGF-1R-initiated signaling, subsequently conferring Herceptin resistance in HER2-positive breast cancer.

## Discussion

While the importance of IGF-1R-initiated signaling in Herceptin resistance has been well documented, the precise mechanism leading to the signaling activation remains elusive. We discover a negative feedback inhibition of IGF2/IGF-1R/IRS1 signaling in HER2-positive breast cancer cells to maintain basic cell survival and proliferation. However, this negative feedback inhibition is disrupted due to dysregulation of the PPP3CB-FOXO3a-miRNA axis in HER2-positive breast cancer resistant to Herceptin (Supplementary Fig. 7f).

Our findings provide several insights into our understanding of the molecular basis of IGF2/IGF-1R/IRS1 signaling activation in Herceptin-resistant breast cancer. First, Herceptin-resistant breast cancer cells over-produced IGF2, triggering autocrine activation

**Fig. 6 Src-mediated STAT6/HDAC1 signaling decreased PPP3CB expression in Herceptin-resistant cells. a** The expression of PPP3CB, p-STAT6, STAT6, p-Src, Src, and β-actin in the indicated cells was analyzed by western blot assays (left). The expression levels of *PPP3CB* mRNA were measured by qRT-PCR (right), ****$p < 0.0001$. **b** Pool2 or HR20 cells with PPP3CB overexpression were treated with Herceptin for 72 h (top). SKBR3 or BT474 cells with specific knockdown of PPP3CB were treated with Herceptin for 72 h (bottom). Cell viability was evaluated by MTS assays, SKBR3 sh-1#: **$p = 0.008$, **$p = 0.0013$, ***$p = 0.0002$, sh-2#: **$p = 0.0019$, ****$p < 0.0001$, BT474 sh-1#: **$p = 0.0038$, ***$p = 0.0002$, ***$p = 0.0001$, sh-2#: ***$p = 0.0003$, ***$p = 0.0001$, ****$p < 0.0001$. **c** Pool2 or HR20 cells were transfected with control vector (Control) or PPP3CB-overexpressing vector (PPP3CB). The expression of PPP3CB, p-FOXO3a, FOXO3a, IRS1, and β-actin was examined by western blot assays (left). IGF2 levels in the CM were detected by ELISA (right). **d** The expression levels of miR-128-3p, miR-30a-5p, and miR-193a-5p in PPP3CB-overexpressing pool2 or HR20 cells were measured by qRT-PCR, ****$p < 0.0001$. **e** Pool2 or HR20 cells were treated with vehicle (Mock) or entinostat (1 μM, Ent.) for 48 h. The expression of PPP3CB, p-FOXO3a, FOXO3a, IRS1, and β-actin was examined by western blots. **f, g** Pool2 or HR20 cells were transfected with control shRNA (Con or sh-Con) or specific *STAT6* shRNAs (sh-2# or sh-4#). The expression of STAT6, PPP3CB, p-FOXO3a, FOXO3a, IRS1, and β-actin was examined by western blots (**f**); the enrichment of HDAC1 at *PPP3CB* promoter was determined by ChIP-qPCR (**g**), ****$p < 0.0001$. **h** Total protein extracts of inducated cells were subjected to IP using an anti-HDAC1 antibody or control IgG, followed by western blot analysis of HDAC1 or p-STAT6. **i** Pool2 or HR20 cells treated with vehicle (Mock) or SU6656 (10 μM) were examined by western blot analysis. $n = 3$ biological independent samples (**a**, **b**, **c**, **d**, **g**). Data are presented as mean values ± SEM (**a**, **b**, **c**, **d**, **g**). Statistical significance was determined by a two-tailed Student's *t*-test (**a**, **b**, **c**, **d**, **g**). All data are provided in the Source Data.

of the IGF-1R signaling, which required IRS1 expression. In support of our data, it has been shown that overexpression of IRS1 causes cell transformation and constitutively active IRS1 promotes tumor growth in various human cancers[34,35]. IRS1 acts as a key mediator of resistance to the inhibitors of EGFR, mTOR, and mutant *B-RAF* in human cancers[36,37]. Next, we found that reduction of several FOXO3a-driven miRNAs played a crucial role in upregulation of both IGF2 and IRS1 in Herceptin-resistant breast cancer cells. Our findings are in agreement with recent reports showing that the *IRS1*-targeting miR-128-3p and miR-30a-5p function as tumor suppressors[38–42]. While the increase of IGF2 in Herceptin resistant cells was attributed to the decrease of miR-193a-5p via inactivation of FOXO3a, this miRNA was shown to be involved in tumorigenesis of endometrial carcinoma through directly targeting YY1[43]. Finally, downregulation of PPP3CB seemed to be pivotal for the increase of p-FOXO3a, lifting FOXO3a-miRNAs axis-controlled expression of both IGF2 and IRS1 in Herceptin-resistant cells. These innovative findings suggested that in addition to PI-3K/Akt signaling, reduced PPP3CB could also increase p-FOXO3a, thereby suppressing its transcription activity. Further studies showed that Src kinase-mediated phosphorylation of STAT6 (p-STAT6) worked cooperatively with HDAC1 to inhibit *PPP3CB* gene transcription. These data not only confirmed our previous findings that IGF-1R-initiated signaling mainly activated Src kinase in Herceptin-resistant cells[24,44], but also emphasized the importance of Src activation in the development of Herceptin resistance[45,46].

Our studies have limitations and their interpretations should be careful. The majority of our data were obtained from in vitro studies using Herceptin-resistant and -sensitive cell lines. Additional in vivo evidence are needed to corroborate the importance of IGF2 and IRS1 upregulation in the development of Herceptin resistance. In addition, we only used a small number of clinical samples. Although the results of correlation analysis seemed to be in line with our conclusions, evaluations with more clinical cases should be performed to provide further support. Nonetheless, we believe that our findings have significant clinical implications. Currently, we do not have validated biomarkers that can predict which HER2-positive breast cancer patients will benefit from Herceptin treatments[7,12]. Our data suggest that IGF2 levels in patients' serum have potential to be developed as a useful biomarker predictive for Herceptin sensitivity. From the therapeutic point of view, our studies may facilitate a rational design of effective treatment strategies to abrogate the resistance to Herceptin. Src inhibition would be effective to overcome Herceptin resistance. This idea has been tested and positive results have

been obtained[46,47]. Our data suggest that the class I HDAC inhibitor entinostat likely holds antitumor activity against HER2-positive breast cancers that are resistant to Herceptin. We previously reported that entinostat mainly inhibited HDAC1 in HER2-positive breast cancer cells[33] and exhibited potent anti-proliferative/anti-survival effects on Herceptin-resistant breast cancer cells[48]. In fact, entinostat exerts profound antitumor activity in various human cancers, including breast cancer[49–51]. While entinostat is actively being tested in clinical trials of cancer patients, further investigations are warranted to determine entinostat's therapeutic potential against HER2-positive breast cancer refractory to Herceptin. Additionally, targeting IGF2 will be effective to overcome Herceptin resistance in cases in which IGF-IR signaling activation is due to the presence of IGF2. One approach is to utilize an IGF2 neutralization Ab that prevents IGF2 binding with its receptors. Several studies have revealed that IGF-IR blockade with Abs results in significant growth inhibition of human cancer cells of breast, renal, pancreas, lung, and colon in vitro and in vivo[52–54]. To date, there is no study showing whether an IGF2 neutralization Ab may abrogate Herceptin resistance in breast cancer. Xentuzumab (BI836845) is a humanized IgG1 monoclonal Ab targeting both IGF1 and IGF2[55]. This Ab neutralizes the ligands and blocks both IGF-1R and insulin receptor (IR) signaling[55,56]. It has been shown to inhibit tumor growth in mouse models[56,57], and is currently under clinical trials of cancer patients, including those with metastatic breast cancer (NCT02123823). We are in the process testing the Ab's therapeutic potential against Herceptin-resistant breast cancer.

In summary, we discover a FOXO3a-miRNA negative feedback inhibition loop to control the IGF2/IGF-1R/IRS1 signaling in HER2-positive breast cancer cells sensitive to Herceptin. In Herceptin-resistant cells, however, this negative feedback inhibition is disrupted via reduction of PPP3CB due to transcriptional repression by p-STAT6/HDAC1 complex. Our studies not only improve our understanding of the molecular mechanism through which IGF-1R signaling activation leads to Herceptin resistance, they also provide new avenues to identify useful biomarkers predictive for Herceptin efficacy and facilitate a rational design of effective strategies to overcome Herceptin resistance.

## Methods

**Reagents and antibodies.** Recombinant human IGF2 (rhIGF2) was from R&D Systems (Minneapolis, MN, USA). Rapamycin, Entinostat, SU6656, and WAY-600 were from Selleck Chemicals (Houston, TX, USA). Cantharidic acid, Cypermethrin and BN-82002 were from Merck (Darmstadt, Germany), Endothall, RK-682, Deltamethrin, RWJ-60475, Tyrphostin 8, CinnGel, and BML-260 were from Enzo

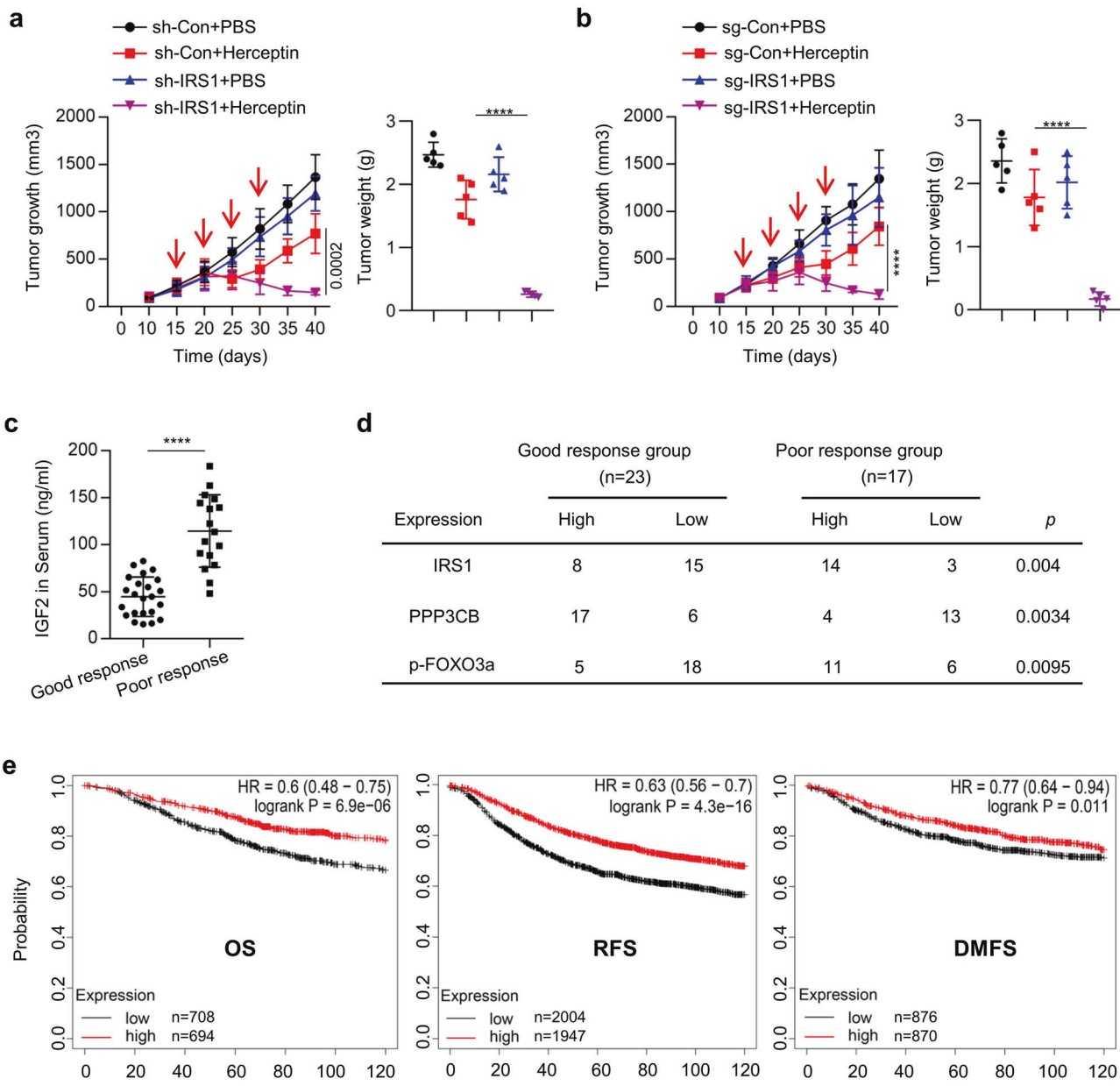

**Fig. 7 PPP3CB/FOXO3a/IRS1 signaling contributed to poor response to Herceptin in vivo. a** Pool2/sh-Con or pool2/sh-IRS1 cells were subcutaneously inoculated into the armpit of female Balb/C athymic nude mice to generate tumor xenografts. When the tumor size reached ~100 mm³, the mice were randomly grouped and received intraperitoneal (i.p.) injection of PBS or Herceptin (10 mg/kg) ($n = 5$) once every 5 days. The tumor growth was measured every 5 days (left). At the experimental endpoint, the tumor weights were measured (right), ****$p < 0.0001$. **b** Pool2 cells with control sgRNA vector (sg-Con) or *IRS1* gene specific sgRNA (sg-IRS1) were subcutaneously inoculated into the armpit of female Balb/C athymic nude mice to generate xenograft tumors. When the tumor sizes reached ~100 mm³, the mice were randomly grouped and received i.p. injection of Herceptin (10 mg/kg) or PBS ($n = 5$) once every 5 days. The tumor growth was measured every 5 days (left). At the experimental endpoint, the tumor weights were measured (right), ****$p < 0.0001$. **c, d** The serum and tumor tissues were obtained from 23 HER2-positive breast cancer patients with a good response to Herceptin-containing treatments and 17 HER2-positive breast cancer patients with a poor response to Herceptin-containing treatments. The levels of IGF2 in the serum were detected by ELISA (**c**), ****$p < 0.0001$; The expression of IRS1, PPP3CB, and p-FOXO3a in the tumors was examined by Immunohistochemistry (IHC) assays (**d**). Data are presented as mean values ± SEM (**c**). Statistical significance was determined by a two-tailed Student's *t*-test (**a–c**). **e** Kaplan–Meier analyses of overall survival (OS), relapse-free survival (RFS), and distant metastasis-free survival (DMFS) in breast cancer patients with low versus high expression of *PPP3CB*. The survival curves were shown with the hazard ratio with 95% confidence intervals and logrank *P*-values.

Biochemicals (New York, NY, USA). Primary antibodies used in western blot assays: IRS1 (#2382, 1:1000), IGF-1R (#3018, 1:1000), p-IGF-1R(Tyr1131)/IR (Tyr1146) (#80732, 1:1000), Akt (#4691, 1:1000), p-Akt(Thr308) (#13038, 1:1000), p-Akt(Ser473) (#4060, 1:1000), S6K (#2708, 1:1000), p-S6K(Thr389) (#97596, 1:1000), FOXO3a (#12829, 1:1000), p-FOXO3a(Ser253) (#9466, 1:1000), STAT6 (#5397, 1:1000), p-STAT6(Tyr641) (#56554, 1:1000), Src (#2109, 1:1000), p-Src (Tyr416) (#59548, 1:1000), and HDAC1 (#34589, 1:1000) were from Cell Signaling

Technology (Danvers, MA, USA). Primary antibodies used in western blot assays: PPP3CB (#HPA008823, 1:1000), PPP3CA (#WH0005530M3, 1:1000), PPP3CC (#SAB1409493, 1:1000), PPP3R1 (#WH0005534M1, 1:1000), PPP3R2 (#SAB1406289, 1:1000), and β-actin (#A5316, 1:3000) were from Sigma (St. Louis, MO, USA). HRP-conjugated secondary antibodies (Goat anti-Rabbit IgG, #31460, 1:5000 and Goat anti-Mouse IgG, #31430, 1:5000) were from Thermo Scientific (Waltham, MA, USA).

**Cells and cell culture**. Human HER2-positive breast cancer cell lines SKBR3 and BT474 were obtained from the American Type Culture Collection (Manassas, VA, USA) and maintained in DMEM medium containing 10% fetal bovine serum (Invitrogen, Carlsbad, CA, USA). Herceptin-resistant sublines pool2 and HR20 were derived from SKBR3 and BT474 cells, respectively[24]. Cells were authenticated using Short Tandem Repeat (STR) analysis with PowerPlex® 18D System from Promega (Madison, WI, USA). Cells were free of mycoplasma contamination, determined by the MycoAlert™ Mycoplasma Detection Kit (Lonza Group Ltd., Basel, Switzerland) every three months.

**Cell transfection**. To establish transfectants with gene knockdown or over-expression, cells were transfected with psi-LVRU6GP vectors containing specific shRNAs or pLEX-MCS vectors containing gene cDNA constructs, respectively, and selected with puromycin (2 μg/ml). To overexpress or inhibit a miRNA, cells were transfected with the miRNA mimic or inhibitor (Ribobio, Guangzhou, China), respectively, using Lipofectamin 3000 (Invitrogen, Carlsbad, CA, USA).

**Clinical analyses**. The clinical samples were collected at the Affiliated Cancer Hospital and Institute of Guangzhou Medical University. All samples were collected with informed consent from the patients and all examining procedures were performed with the approval of the Internal Review and Ethics Boards (IRB) of the hospital. Our study is compliant with the 'Guidance of the Ministry of Science and Technology (MOST) for the Review and Approval of Human Genetic Resources' and it has been formally approved for the export of human genetic material or data from China. Serum samples and tumor tissues were obtained from 40 HER2-positive breast cancer patients treated with the neoadjuvant Herceptin-containing regimen TAC (Docetaxel, Doxorubicin and Cyclophosphamide). Drs. Ni Qiu and Hongsheng Li, who are physician scientists and our collaborators in the current study, assessed the patients' responses. Good response was defined as CR (Complete Response) with disappearance of all target lesions or PR (Partial Response) with at least 30% decrease in the sum of diameters of target lesions after the neoadjuvant therapy including Herceptin. Poor response was defined as SD (Stable Disease) without sufficient shrinkage to quality for PR or PD (Progressive Disease) with at least 20% increase in the sum of diameters of target lesions.

**Tumor xenograft model**. The animal studies were approved by the Institutional Animal Care and Use Committee (IACUC) of Guangzhou Medical University. Standard animal care and laboratory guidelines were followed according to the IACUC protocol. Mice were housed in individually ventilated cages, under the standard room temperature (22 °C) and humidity (55%), 12/12 light/dark cycle. Human breast cancer cells were inoculated subcutaneously into the armpit of female Balb/C athymic nude mice (Guangdong Medical Laboratory Animal Center, Guangzhou, China) to generate xenograft tumors. When the tumor sizes reached ~100 mm³, the mice were randomly grouped and injected intraperitoneally (i.p.) with Herceptin (10 mg/kg) or PBS as control ($n = 5$). The treatment was administered every 5 days for four cycles. The tumor growth was measured every 5 days. The wet weight of tumors was recorded after excised at the experimental endpoint.

**Western blot assays**. Total protein was extracted from cells using RIPA buffer (Thermo Scientific) in the presence of Halt Protease and Phosphatase Inhibitor Cocktail (Pierce Chemical, Dallas, Texas, USA). Protein concentration was measured using a BCA Protein Assay Kit (Thermo Scientific). Equivalent amounts of protein were mixed with 5×Lane Marker Reducing Sample Buffer (Thermo Scientific), and resolved by a SDS–polyacrylamide gel and then transferred onto Immobilon-P Transfer Membrane (Millipore, Burlington Mass, USA). The membranes were blocked with 5% non-fat milk in Tris-buffered saline and then incubated with a primary antibody followed by secondary antibody. The signal was detected using enhanced chemiluminescence western blot detection kit (Millipore). Densitometry analyses of the signals were measured with ImageJ software (v2.0.0). Uncropped blots are available in the file of Source Data.

**Total RNA and miRNA Isolation and RT-qPCR**. Total RNA was isolated from cells with a RNA isolation Kit (QIAGEN, Hilden, Germany) according to the manufacturer's instructions. First-strand cDNA synthesis was synthesized with the first-strand synthesis system (Thermo Scientific). Real-time PCR was carried out using an ABI 7500 with SYBR Green detection (Applied Biosystems, Foster City, CA, USA) by the CFX96 Real-Time PCR System (Bio-Rad iQ5 program). GAPDH was used as an internal control. miRNAs were isolated from cultured cells and purified with the miRCURY RNA Isolation Kit (Exiqon, Vedbaek, Denmark). cDNA was generated with the All-in-One™ miRNA First-Strand cDNA Synthesis Kit (GeneCopoeia, Guangzhou, China), and quantitative real-time PCR (qRT-PCR) was performed by using the All-in-One™ miRNA qPCR Kit (GeneCopoeia) according to the manufacturer's instructions. The miRNA sequence-specific RT-PCR primers and the endogenous control RNU6 were purchased from GeneCopoeia. The relative quantitative expression was calculated by normalizing the results with RNU6. Primer sequences are provided in Supplementary Table 1.

**ELISA analysis**. Serum and the conditioned medium (CM) of cultured cells were analyzed to measure IGF1 and IGF2 levels using ELISA kits (R&D Systems, Minneapolis, MN, USA). The assays were performed in triplicates according to the manufacturer's instructions.

**Cell proliferation assay**. To determine the sensitivity of cells to Herceptin, cell proliferation (MTS) assays were performed using a CellTiter 96® AQueous One Solution Cell Proliferation Assay kit (Promega, Madison, WI, USA) according to the manufacturer's instructions. Briefly, cells were seeded onto 96-well plates at 2000 cells/well (0.20 ml/well) with different concentrations of Herceptin. The cell viability was determined on day 2 by incubation with MTS (0.02 ml/well). After 2 h of incubation, the absorbance at 490 nm representing cell viability was recorded for each well on a BioTek Synergy 2 system, and the cell viability was calculated for each time point.

**Luciferase reporter assay**. For miRNA luciferase reporter assays: DNA sequences from IRS1 3′UTR or IGF2 3′UTR were cloned into pMir-Report plasmid downstream of firefly luciferase reporter gene. Cells were seeded onto 96 well-plates and co-transfected with pMir-Report luciferase vector, pRL-TK Renilla luciferase vector and related miRNA inhibitors or mimics using Lipofectamine 3000 (Invitrogen). For promoter activity assays: potential promoters were cloned into the pGL4-reporter vector upstream of the luciferase gene. Cells were seeded onto 96-well plates and co-transfected with the pGL4-reporter vector and the pRL-TK Renilla luciferase vector with or without FOXO3a-shRNA using Lipofectamine 3000 (Invitrogen). After transfection of 48 h, luciferase activity was determined using a Dual-Luciferase Reporter Assay System (Promega) on the BioTek Synergy 2. The Renilla luciferase activity was used as internal control and the firefly luciferase activity was calculated as the mean ± SD after being normalized by Renilla luciferase activity.

**ChIP-qPCR analysis**. The ChIP-qPCR assays were performed using EZ-CHIP™ chromatin immunoprecipitation kit (Merck Millipore). Briefly, chromatin proteins were cross-linked to DNA by addition of formaldehyde to the culture medium with a final concentration of 1%. After 10 min incubation at room temperature, the cells were washed and scraped off in ice-cold PBS containing Protease Inhibitor Cocktail II. Cells were pelleted and then resuspended in lysis buffer containing Protease Inhibitor Cocktail II. The resulting lysates were subjected to sonication to reduce the size of DNA to approximately 200-1000 base pairs in length. The samples were centrifuged to remove cell debris and diluted ten-fold in ChIP dilution buffer containing Protease Inhibitor Cocktail II. A 5 μl sample of the supernatant was retained as "Input" and stored at 4 °C. Then, 5 μg of an Ab were added to the chromatin solution and incubated overnight at 4 °C with rotation. After Ab incubation, protein G-agarose (Amersham Biosciences, Piscataway, NJ, USA) was added and the samples were incubated at 4 °C with rotation for additional 2 h. The protein/DNA complexes were washed with Wash Buffer four times and eluted with ChIP Elution Buffer. Cross-links were then reversed to free DNA by the addition of 5 M NaCl and incubation at 65 °C for 4 h. DNA were purified according to the manufacturer's instructions. 0.2 μl of DNA from each group was used as a template for PCR. qRT-PCR was carried out according to the standard protocol. Details of primer sequences for ChIP-qPCR are indicated in Supplementary Table 2. The results were calculated by normalizing to the positive control, and relative quantization values were calculated using % input = 2^(−ΔCt [(Ct [14-3-3σ] − (Ct [input]]) method.

**CRISPR-Cas9 gene deletion**. Gene deletion in breast cancer cells were generated with lentivirus-mediated CRISPR-Cas9 technology (Genechem, Shanghai, China). SgRNA sequences targeting human IRS1 were as follows: 1#, TGGCTTCTCGGACGTGCGCA; 2#, TGGGCCGTTCTGCCGTGACG. SgRNA sequences targeting human FOXO3a were: 1#, ACTGCCACGGCTGACTGATA; 2#, GGCGACAGCAACAGCTCTGC. Cells were first infected with lentivirus containing constructs encoding Cas9 (Lenti-CAS9-puro). After selection with puromycin (2 μg/mL), cells were then infected with lentivirus containing sgRNA constructs (GV371-EGFP). After selection by sorting EGFP positive cells, the efficiency of gene knockout was examined and subsequent experiments were performed.

**Co-immunoprecipitation**. Cells were lysed with modified TNE buffer (50 mM Tris [pH 8.0], 150 mM NaCl, 1% Nonidet P-40 [NP-40], 10 mM sodium fluoride, 10 mM sodium pyrophosphate, 2 mM EDTA) supplemented with 1 mg/L leupeptin, 1 mg/L aprotinin, and 1 mM sodium orthovanadate (Na₃VO₄). The immunoprecipitations were performed overnight at 4 °C with a specific Ab or IgG (negative control). The immunoprecipitates were then incubated for 2 h with protein G-agarose (Amersham Biosciences). The reaction products were washed with lysis buffer, and the immune complexes were resolved by SDS-PAGE followed by western blot assays.

**Immunohistochemistry (IHC) assay**. The tissues after formalin-fixed and paraffin-embedded were cut into 4-μm sections. The specimens were deparaffinized in xylene and rehydrated using a series of graded alcohols after being

dried at 62 °C for 2 h. The slides were then treated with 3% hydrogen peroxide in methanol for 15 min. To exhaust endogenous peroxidase activity, the antigen was retrieved in 0.01 M sodium cirate buffer (pH 6.0) using a microwave oven. After 1 h of preincubation in 10% goat serum, the specimens were incubated with a primary Ab at 4 °C overnight. The slides were treated with a horseradish peroxidase detection system (DAKO, Glostrup, Denmark) according to the manufacturer's instruction. Two independent individuals evaluated the slides. The intensity of immunostaining was taken into consideration when analyzing the data.

**Statistical analysis**. All data were presented as the mean values ± SEM. Statistical analyses were performed using SPSS version 16.0 and GraphPad Prism 7. A chi-square test was used to analyze the relationship between genes expression levels. Student's $t$-tests were performed to calculate the $p$ values, and $p < 0.05$ was considered statistically significant. The Kaplan–Meier survival analyses and the hazard ratio with 95% confidence intervals compared the two patient cohorts with high or low *PPP3CB* expression and logrank P values were calculated.

**Reporting summary**. Further information on research design is available in the Nature Research Reporting Summary linked to this article.

## Data availability

All data supporting this study are available within this article, the Supplementary Information file, and the Source data as indicated in the Reporting Summary of this article. Source data are provided with this paper. The patient survival curves were generated using the Kaplan–Meier plotter (https://kmplot.com/analysis/index.php?p=service&cancer=breast).

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

## Acknowledgements

The authors are grateful to Dr. Francisco J. Esteva (MD Anderson Cancer Center) for providing SKBR3-pool2 cells and to Dr. XiaoHe Yang (North Carolina Central University) for providing BT474-HR20 cells. This study was supported in part by the grants (81402196, 81672616, and 81872197) from the National Natural Science Foundation of China (NSFC). It was also supported in part by Guangdong Natural Science Funds for Distinguished Young Scholars (2016A030306003), Guangdong Natural Science Funds (2017A030313867), Guangdong Special Support Program (2017TQ04R809), Guangzhou Key Medical Discipline Construction Project Fund, and Science and Technology Program of Guangzhou (201710010100).

## Author contributions

L. Luo, Z.H., B.L., and G.Z. designed the study. Z.Z., Z.H., B.L., and G.Z. provided conceptual advice and supervised the study. L. Luo, Z.Z., N.Q., L. Ling, X.J., and H. Lyu developed research methodology. L. Luo, Z.Z., N.Q., H. Li, Y.S., J.L., L. Ling, and X.J. performed the experiments, collected and analyzed the experimental data. L. Luo, Z.Z., H. Liu, and Z.H. provided administrative, technical, and material support. Z.H., B.L., and G.Z. performed literature review, wrote and finalized the manuscript with the contributions of other authors.

## Competing interests

The authors declare no competing interests.
