## [Peer Review File · Nature Communications]

Reviewers' Comments:

Reviewer #1:

Remarks to the Author:

The manuscript discovered a novel mechanism of Herceptin resistance in HER2 positive breast cancer. Their study found that FOXO3a regulates specific miRNAs to control the IGF2/IGF1R/IRS1 signaling in Herceptin-sensitive cells. PPP3CB, a subunit of the serine/threonine-protein phosphatase 2B, can activate the transcriptional activity of FOXO3a. In Herceptin-resistant cells, low level of PPP3CB enhanced p-FOXO3a, inhibited the expression of specific miRNAs, and disrupted the negative feedback loop formed by FOXO3a and miRNAs.

Overall the findings are interesting and novel. The authors should address the following issues:

1. In Fig 3e, they found that FOXO3a binds to the promoter region of specific microRNAs, then promotes the expression of these microRNAs. Since PPP3CB promotes transcriptional activity of FOXO3a, it would be interesting to examine whether knockdown of PPP3CB will change the binding of FOXO3a to the promoter of these microRNAs.
2. In Fig 3, they identified that both miR-128-3p and miR-30a-5p target IRS1. The targeting sequence of IRS1 should be showed.
3. In Fig 4a, IGF2 induced the expression of PPP3CB, and Rapamycin treatment abolished the induction of PPP3CB. It's not clear how mTOR regulates IGF2 induced PPP3CB expression. Has it been reported before?
4. In Fig 5f, WAY-600 treatment reduced p-FOXO3a, but it's not clear whether it may also change the level of PPP3CB in resistant cells, so PPP3CB should be checked by western blot analysis.
5. The study showed that PPP3CB is an important regulator of FOXO3a. In Fig 6a, they detected higher expression levels of PPP3CB in sensitive cells than resistant cells, so some experiments can be designed to check whether sensitive cells expressing PPP3CB knockdown will become resistant to Herceptin and overexpression of PPP3CB will sensitize resistant cells to Herceptin.
6. For mouse model experiment in Fig 7, expression level of p-FOXO3a, IRS1 and p-Akt should be tested by immunohistochemistry analysis.

Reviewer #2:

Remarks to the Author:

In this manuscript the authors report that resistance to Herceptin treatment of HER2+ breast cancers could be due to activation of the IGF2/IRS1 signal because of disruption of a negative loop between FOXO3a-miR-128 and miR-30

There are several technical problems associated with this study

General comments

Supplementary figures all lack statistical tests, this issue should be addressed. It is unclear why main figures deserve statistic but not supplementary material.

Western blotting mostly show signals in saturation, therefore especially for loading controls such as b-actin, is difficult to assess whether the levels of the proteins loaded is truly equal amongst the various lanes. This is true for most of the western blottings shown. The authors should show western blotting images containing signal after lower exposures, when signals have not yet reached saturation. This is very important and would permit to better assess the quality of the data.

Another issue is that the authors only show a single western blotting for each experiment in all cases. It would be more appropriate to show a representative western blotting as well as graphs of densitometric scanning ratios between target and loading controls derived from at least three independent experiments, thoroughly. In addition to this, statistical test should be performed to provide significance evidence of the changes shown.

Specific comments

Experiments in supplementary figure 1 lacks statistic. Also, in supplementary figure 1b and c it would be more appropriate to show data in combined box/whisker plot instead of barplot to have a

better idea of the experimental variations.

The authors have used shRNA against IRS1 to test its role in Herceptin resistance and mTOR activation. It would be appropriate to also use CRISPR/CAS9 against IRS1 in these cell lines as an additional tool to test reproducibility of IRS1 resistance phenotypes

It is generally unclear how many times the western blotting experiments have been done. It would be appropriate to also provide densitometric scan of the bands and to plot the average of these values divided by actin control accompanied to standard deviations and statistical tests.

Figure 2c, in addition of FOXO3a knockdown experiments, CRISPR/CAS9 should be used to test reproducibility of FOXO3a-mediated effects.

Supplementary figure 3a. The authors should also measure miRNAs that do not change as control in addition to miR-128-3p and miR-30a-5p.

Figure 3a, b-actin signal is in saturation, difficult to assess whether inhibition of miR-128-3p and miR-30a-5p really increases IRS1 expression. Representative western blotting with lower exposure intensity should be shown and average of three independent experiment and p-values should be plotted.

Also, how can the authors can explain that in figure 3b there is not increase in IRS1 protein levels upon miRNA inhibition? (For example in figure 3b comparing lane 2, with lane 5, 8 and 11 does not seem to be any difference in IRS1 expression). Additionally, what happens to IRS1 mRNA upon miRNA ectopic modification? What happen in cells where the genomic loci expressing miR-30 and miR-128-3p are removed by CRISPR/CAS9?

Figure 4b. In addition to miR-128-3p and miR-30a-5p control miRNAs that do not change should be shown.

Supplementary figure 5d, why miR-126 expression is shown and why changes expression of this miRNA is similar to miR-128/30b and 193? Maybe a general process regulating general miRNA biogenesis could explain these effects here? Is there any miRNA that do not change in these conditions?

Reviewer #3:

Remarks to the Author:

In the present article Luo L et al, describes the role of FOXO3a- miRNA in the control of IGF2/IGFR1/IRS1 axis in relation to resistance to trastuzumab. The article is well developed, and mechanistically is well executed, including in vitro and in vivo models.

However, the major limitation is the fact that the novelty of the findings described are not extremely new. The role of the IGF2/IGFR1/IRS1 axis in resistance to trastuzumab is well described and documented.

Responses to Reviewer #1:

The manuscript discovered a novel mechanism of Herceptin resistance

Overall the findings are interesting and novel. The authors should address the following issues:

Response: We thank the reviewer for his/her generous comments of research work.

1. In Fig 3e, they found that FOXO3a binds to the promoter region of specific microRNAs, then promotes the expression of these microRNAs. Since PPP3CB promotes transcriptional activity of FOXO3a, it would be interesting to examine whether knockdown of PPP3CB will change the binding of FOXO3a to the promoter of these microRNAs.

Response: We appreciate the reviewer for this constructive criticism. According to the suggestion, we performed new experiments with specific shRNAs to downregulate PPP3CB expression. Our data revealed that specific knockdown of PPP3CB not only dramatically increased the levels of p-FOXO3a in both SKBR3 and BT474 cells treated with high concentration of rhIGF2 (80 ng/ml), and it also significantly abolished rhIGF2 (80 ng/ml)-mediated enrichment of FOXO3a at the promoters of miR-128-3p and miR-30a-5p. These new findings are now shown in figure 4e and 4f, respectively, in the revised manuscript.

2. In Fig 3, they identified that both miR-128-3p and miR-30a-5p target IRS1. The targeting sequence of IRS1 should be showed.

Response: The targeting sequences of *IRS1* by the two miRNAs have now been provided in the Supplementary figure 3a.

3. In Fig 4a, IGF2 induced the expression of PPP3CB, and Rapamycin treatment abolished the induction of PPP3CB. It's not clear how mTOR regulates IGF2 induced PPP3CB expression. Has it been reported before?

Response: We thank the reviewer for this stimulating discussion. In the present study, our data indicated that high concentration of rhIGF2 (80 ng/ml) induced expression of PPP3CB via mTOR at protein, but not mRNA level, implying a translational regulation of PPP3CB by mTOR. To the best of our knowledge, this finding is novel and it has not been reported. Nonetheless, the underlying mechanism of mTOR-mediated translational regulation of PPP3CB remains unknown, and will be investigated in our future studies.

4. In Fig 5f, WAY-600 treatment reduced p-FOXO3a, but it's not clear whether it may also change the level of PPP3CB in resistant cells, so PPP3CB should be checked by western blot analysis.

Response: We examined PPP3CB expression by western blots, and did not detect significant changes of PPP3CB level upon WAY-600 treatment in the resistant cells. These observations may be a reflection that the Herceptin-sensitive and -resistant cells exhibit distinct mechanisms regulating PPP3CB expression by IGF-1R signaling. In the sensitive cells, mTOR modulated expression of PPP3CB at translational level but not transcriptional level, to influence IGF2-induced changes in p-FOXO3a, whereas PPP3CB was downregulated by STAT6/HDAC1

complex at transcriptional level in the resistant cells. Thus, our studies demonstrated that PPP3CB was translationally regulated by mTOR-mediated negative feedback regulation of IGF2/IRS1/mTOR signaling in sensitive cells. However, the STAT6/HDAC1 complex in the resistant cells transcriptionally suppressed PPP3CB, thereby conferring a constitutive activation of IGF2/IRS1/mTOR signaling.

5. The study showed that PPP3CB is an important regulator of FOXO3a. In Fig 6a, they detected higher expression levels of PPP3CB in sensitive cells than resistant cells, so some experiments can be designed to check whether sensitive cells expressing PPP3CB knockdown will become resistant to Herceptin and overexpression of PPP3CB will sensitize resistant cells to Herceptin.

Response: Thank the reviewer for this helpful suggestion. We performed additional experiments and found that ectopic expression of PPP3CB re-sensitized the resistant cells to Herceptin. In contrast, specific knockdown of PPP3CB elicited the sensitive cells becoming resistant to Herceptin. These data are now shown in Fig. 6b and supplementary figure 6a in the revision.

6. For mouse model experiment in Fig 7, expression level of p-FOXO3a, IRS1 and p-Akt should be tested by immunohistochemistry analysis.

Response: We performed immunohistochemistry assays to examine the levels of p-FOXO3a, IRS1, and p-Akt in the tumor tissues obtained from our animal experiments. IRS1 knockdown or deletion markedly decreased the levels of p-Akt and p-FOXO3a *in vivo* (Supplementary figure 7c), confirming that p-Akt/p-FOXO3a acting as the downstream of IRS1 in resistant cells.

Responses to Reviewer #2:

In this manuscript the authors report that resistance to Herceptin treatment of HER2+ breast cancers could be due to activation of the IGF2/IRS1 signal because of disruption of a negative loop between FOXO3a-miR-128 and miR-30. There are several technical problems associated with this study.

General comments

1. Supplementary figures all lack statistical tests, this issue should be addressed. It is unclear why main figures deserve statistic but not supplementary material.

Response: We appreciate the reviewer for his/her helpful comments. Statistical analyses on majority of the supplementary data have been performed and are now included in the revised Supplementary figures.

2. Western blotting mostly show signals in saturation, therefore especially for loading controls such as b-actin, is difficult to assess whether the levels of the proteins loaded is truly equal amongst the various lanes. This is true for most of the western blottings shown. The authors should show western blotting images containing signal after lower exposures, when signals have not yet reached saturation. This is very important and would permit to better assess the quality of the data.

Response: Thank you for the kind suggestions. We repeated a number of our western blot assays, and took a shorter exposure time during film development. Some of new data with lower exposures are now included in the revised manuscript.

3. Another issue is that the authors only show a single western blotting for each experiment in all cases. It would be more appropriate to show a representative western blotting as well as graphs of densitometric scanning ratios between target and loading controls derived from at least three independent experiments, thoroughly. In addition to this, statistical test should be performed to provide significance evidence of the changes shown.

Response: All of the western blot assays were repeated at least three times, and sometimes by two independent lab people. We fully understand the reviewer's concern about the quantification issue of our western blotting. We have performed densitometric analysis on the western blot gels. Since we have a huge amount of data and each figure contains multiple panels, the space limitation becomes problematic. Thus, we have organized the data of our densitometric analysis on the key proteins into an Excel file, which is now shown as "Source Data" in the revised manuscript. I hope that this arrangement can satisfy the reviewer.

Specific comments

4. Experiments in supplementary figure 1 lacks statistic. Also, in supplementary figure 1b and c it would be more appropriate to show data in combined box/whisker plot instead of barplot to have a better idea of the experimental variations.

Response: Supplementary figure 1 with statistical analysis is now shown in the revision. We have also replaced the barplots (original supplementary figure 1b and c) with box and whisker or scatter plots.

5. The authors have used shRNA against IRS1 to test its role in Herceptin resistance and mTOR activation. It would be appropriate to also use CRISPR/CAS9 against IRS1 in these cell lines as an additional tool to test reproducibility of IRS1 resistance phenotypes.

Response: It was greatly appreciated for this constructive suggestion. We utilized CRISPR/Cas9 gene editing technology to knockout *IRS1* and performed additional assays, including cell viability examination upon Herceptin treatment, western blot detection, and *in vivo* animal experiments. We are extremely happy that our new data (Fig. 1d, Fig. 1e, and Fig. 7b in the revised manuscript) are able to confirm the reproducibility of IRS1-mediated resistance phenotypes.

6. It is generally unclear how many times the western blotting experiments have been done. It would be appropriate to also provide densitometric scan of the bands and to plot the average of these values divided by actin control accompanied to standard deviations and statistical tests.

Response: We have done at least three times for each western blot experiment to verify our data. The densitometric analyses of the western blot data regarding the signal ratios between key targets and loading controls are included in the Excel file of "Source Data".

7. Figure 2c, in addition of FOXO3a knockdown experiments, CRISPR/CAS9 should be used to test reproducibility of FOXO3a-mediated effects.

Response: Once again, we performed a number of additional assays upon *FOXO3a* gene deletion via a CRISPR/Cas9 technology. Our studies strongly confirmed FOXO3a-mediated effects. These new data are now shown in Fig. 2d, Fig. S2e, Fig. S2f and Fig. S3e. We have also revised our manuscript accordingly.

8. Supplementary figure 3a. The authors should also measure miRNAs that do not change as control in addition to miR-128-3p and miR-30a-5p.

Response: Thanks for this kind suggestion. In fact, we did examine other miRNAs. The expression levels of miR-191-5p were found no change upon rhIGF2 treatment. Due to space limitation and the layout problem, the data were not shown. We included this information in the revised manuscript.

9. Figure 3a, b-actin signal is in saturation, difficult to assess whether inhibition of miR-128-3p and miR-30a-5p really increases IRS1 expression. Representative western blotting with lower exposure intensity should be shown and average of three independent experiment and p-values should be plotted.

Response: We repeated the experiment for several times and obtained similar results. A lighter exposure of b-actin signal is now provided in the revised figure 3a. Our new data showed a clear increase of IRS1 expression upon inhibition of miR-128-3p and miR-30a-5p. We have also provided the data of densitometry analyses of the signal intensity in “Source Data”.

10. Also, how can the authors can explain that in figure 3b there is not increase in IRS1 protein levels upon miRNA inhibition? (For example in figure 3b comparing lane 2, with lane 5, 8 and 11 does not seem to be any difference in IRS1 expression). Additionally, what happens to IRS1 mRNA upon miRNA ectopic modification? What happen in cells where the genomic loci expressing miR-30 and miR-128-3p are removed by CRISPR/CAS9?

Response: We appreciate the reviewer for his/her thorough evaluation of our data. We regret for our unclear description in the original manuscript. I hope that we have made our point clearer in the revised document. Collectively, our studies demonstrated that low dose of rhIGF2 (10ng/ml) upregulated IRS1 levels via downregulation of miR-128-3p and miR-30a-5p due to inactivation of FOXO3a. The treatment of SKBR3 or BT474 cells with rhIGF2 (10ng/ml) profoundly reduced the expression levels of miR-128-3p and miR-30a-5p (fig. 3c). Because the miRNAs' levels were so low, they no longer responded to the miRNA inhibitors. Thus, it is conceivable to believe that there should be no significant difference in IRS1 levels comparing line 2 with lane 5, 8 and 11.

Additionally, we had tried to remove the genomic loci of miR-30-5p and miR-128-3p by using CRISPR/CAS9 technology. Unfortunately, the experiments failed. Nevertheless, we feel confident about our data obtained from the studies using both mimics and inhibitors, which

strongly confirm the effects of miR-30-5p and miR-128-3p via both positive and negative aspects.

11. Figure 4b. In addition to miR-128-3p and miR-30a-5p control miRNAs that do not change should be shown.

Response: This question is similar as #8. The expression levels of miR-191-5p had no change upon the treatments in both SKBR3 and BT474 cells (data not shown).

12. Supplementary figure 5d, why miR-126 expression is shown and why changes expression of this miRNA is similar to miR-128/30b and 193? Maybe a general process regulating general miRNA biogenesis could explain these effects here? Is there any miRNA that do not change in these conditions?

Response: We regret that this ever happened and apologize for the typo. It should be described as miR-128-3p, not miR-126. We have corrected this error in the revision.

Responses to Reviewer #3:

1. In the present article Luo L et al, describes the role of FOXO3a- miRNA in the control of IGF2/IGFR1/IRS1 axis in relation to resistance to trastuzumab. The article is well developed, and mechanistically is well executed, including in vitro and in vivo models.

Response: We sincerely appreciate the reviewer for his/her positive comments, regarding our mechanistic studies using both *in vitro* and *in vivo* models.

2. However, the major limitation is the fact that the novelty of the findings described are not extremely new. The role of the IGF2/IGFR1/IRS1 axis in resistance to trastuzumab is well described and documented.

Response: We agree with the reviewer that the role of the IGF2/IGF-1R/IRS1 axis in the development of Herceptin resistance is well documented in the literature. Nevertheless, the precise mechanism through which IGF-1R-initiated signaling modulates Herceptin sensitivity remains elusive. Especially, it is not clear whether specific miRNAs and/or any protein phosphatases may involve in the regulation of the IGF2/IGF-1R/IRS1 axis; and it is unknown whether IGF2 may have potential to be developed as a novel biomarker predictive for the treatment response to Herceptin.

In our study, we discover that the transcription factor FOXO3a and several *IGF2*- and *IRS1*-targeting miRNAs form a negative feedback inhibition loop to control the IGF2/IGF-1R/IRS1 signaling in Herceptin sensitive breast cancer cells. In the resistant cells, however, this negative feedback inhibition loop is disrupted. Further studies demonstrate that this disruption is due to the downregulation of PPP3CB, a subunit of the serine/threonine-protein phosphatase 2B, which may function as a tumor suppressor in breast cancer. We believe that our data provide significant insights in the molecular basis of IGF2/IGF-1R/IRS1 signaling axis-mediated Herceptin resistance, they may also facilitate the development of IGF2 as a useful biomarker

predictive for Herceptin efficacy and the rational design of effective therapeutic strategies to overcome the resistance.

Reviewers' Comments:

Reviewer #1:

Remarks to the Author:

The authors have done a good job of addressing my comments. I am satisfied with the revisions that were made, and have no other comments.

Reviewer #2:

Remarks to the Author:

The authors mostly addressed my concerns.

They should only show expression of unchanging miRNA (miR-191-5p) in the appropriate figures. There is space to include these data, 'data not shown' is unacceptable to me.

Responses to Reviewer #1:

The authors have done a good job of addressing my comments. I am satisfied with the revisions that were made, and have no other comments.

Response: We thank the reviewer for his/her generous comments of our revisions.

Responses to Reviewer #2:

The authors mostly addressed my concerns.

They should only show expression of unchanging miRNA (miR-191-5p) in the appropriate figures. There is space to include these data, 'data not shown' is unacceptable to me.

Response: We appreciate the reviewer for his/her helpful suggestions. We have now included miR-191-5p data in the revised supplementary figure 3b and supplementary figure 4b & c. The expression levels of miR-191-5p remained unchanged upon the treatments in both SKBR3 and BT474 cells.